# The rise and global spread of IMP carbapenemases (1996-2023): a genomic epidemiology study

Ben Vezina [1,6], Bhargava Reddy Morampalli [1,6], Hoai-An Nguyen [1], Angela Gomez-Simmonds [2], Anton Y. Peleg [1,3,4] & Nenad Macesic [1,3,5] ✉

Infections caused by carbapenemase-producing organisms are a global health threat. IMP carbapenemases are one of the key drivers of these infections but little is known regarding their global epidemiology. We analyse three decades of $bla_{IMP}$ gene spread using sequence data from 4556 genomes collected between 1996–2023. A total of 52 $bla_{IMP}$ variants were identified across 93 bacterial species. We reconstruct the historical emergence and variant-specific epidemiologies of $bla_{IMP}$ genes and showed how key variants ($bla_{IMP-1}$, $bla_{IMP-4}$, $bla_{IMP-7}$, $bla_{IMP-8}$ and $bla_{IMP-13}$) achieved global endemicity, while $bla_{IMP-26}$ and $bla_{IMP-27}$ became regionally endemic in Southeast Asia and North America, respectively. Dissemination was driven predominantly by horizontal gene transfer facilitated by mobile genetic elements such as class 1 integrons and insertion sequences. These elements mobilised $bla_{IMP}$ genes into 52 distinct plasmid clusters (predominantly IncHI2A, IncN, IncL/M, and IncC), enabling broad inter-species transmission. Despite limited overall cross-source transmission, spillover primarily occurred between human and environmental reservoirs. Structural analysis revealed conserved IMP carbapenemase structure (mean lDDT 0.977) with convergent missense mutations at seven catalytically relevant sites. Our analysis provides a framework for understanding $bla_{IMP}$ dissemination, highlighting their emergence as an important, yet under-recognised, public health threat.

Carbapenemase-producing organisms (CPOs) are a significant threat to global health and have been deemed critical priority pathogens by the World Health Organization[1]. Five key carbapenemase classes (KPC, NDM, OXA, VIM and IMP) cause the majority of global infections[2,3]. Despite being included amongst these, little is known about the global epidemiology of IMP carbapenemases. IMP carbapenemases are metallo-beta-lactamases (MBLs) that were first identified in 1991 in *Pseudomonas aeruginosa* in Japan[4]. To date, 96 different IMP carbapenemase variants have been identified and IMP carbapenemases are now endemic to Asia and Australia[5–10]. In addition, outbreaks of IMP-carrying organisms are increasingly reported across several regions including Europe and the Americas[11–15]. This is a highly concerning development given the paucity of treatment options for infections caused by these organisms, including resistance to novel agents with activity against other MBLs such as cefepime-taniborbactam[16].

¹Department of Infectious Diseases, The Alfred Hospital and School of Translational Medicine, Monash University, Melbourne, VIC, Australia. ²Division of Infectious Diseases, Department of Internal Medicine, UC Davis Health, Sacramento, CA, USA. ³Centre to Impact AMR, Monash University, Clayton, VIC, Australia. ⁴Infection Program, Monash Biomedicine Discovery Institute, Department of Microbiology, Monash University, Clayton, VIC, Australia. ⁵Infection Prevention & Healthcare Epidemiology, Alfred Health, Melbourne, VIC, Australia. ⁶These authors contributed equally: Ben Vezina, Bhargava Reddy Morampalli. ✉e-mail: nenad.macesic1@monash.edu

Carbapenemases disseminate through various mechanisms, including transposon-mediated transfer between plasmids ($bla_{NDM}$), stable association with successful clonal lineages ($bla_{KPC}$), rapid expansion of a single epidemic plasmid across multiple bacterial lineages ($bla_{OXA-48}$), and transient associations involving diverse plasmids and numerous lineages[17,18]. Our prior work indicated that in Australia $bla_{IMP-4}$ spreads both clonally and through horizontal transfer via mobile genetic elements[19]. However, current data on $bla_{IMP}$ dissemination remain limited: most prior studies have focused on a single IMP carbapenemase type and/or a specific geographical region[5,19–23].

We therefore aimed to comprehensively determine the genomic epidemiology of $bla_{IMP}$ carbapenemase genes. Specifically, we dissected the dynamics underlying $bla_{IMP}$ dissemination and evaluated the contributions of genomic factors, outbreak events, structural determinants and One Health-related influences. We analysed all publicly-available $bla_{IMP}$-carrying genomes ($n = 4556$) spanning almost three decades (1996–2023), uncovering global expansion and regional endemicity of diverse $bla_{IMP}$ variants. Collectively, this work creates an atlas of $bla_{IMP}$ carbapenemase genes that highlights their transition from initial endemic foci in the Asia-Pacific region to a worldwide public health threat and emphasises the pressing need for integrated strategies to combat their further spread.

## Results

We identified 4556 genomes (4020 assembled from short- and 536 from long-read sequencing data) isolated globally from 1996–2023 carrying 52 distinct $bla_{IMP}$ variants across 26 bacterial genera (Fig. 1 and Supplementary Data 1). This revealed a remarkable diversity of both $bla_{IMP}$ genes and their bacterial hosts, totalling 4559 $bla_{IMP}$ genes,

with three long-read genomes carrying two $bla_{IMP}$ variants each. $bla_{IMP-4}$ and $bla_{IMP-1}$ were the most frequent variants, found in 1592/4559 (34.9%) and 1155/4559 (25.3%) genomes, respectively (Supplementary Data 2). The most prevalent species included 1053 *Enterobacter hormaechei* (23.1%), 977 *Pseudomonas aeruginosa* (21.4%) and 681 *Klebsiella pneumoniae* (14.8%), together accounting for 59.4% of the dataset. Of the 4556 $bla_{IMP}$-carrying genomes, 728 (16%) carried mobile colistin resistance (*mcr*) genes and 345 (7.6%) carried other carbapenemase genes (Supplementary Data 1). We did not identify any uncatalogued $bla_{IMP}$ variants.

### Tracking the global spread of $bla_{IMP}$ carbapenemase genes

Our dataset enabled us to reconstruct the global expansion of $bla_{IMP}$ genes from origins in the Asia-Pacific region to an increasing number of variants detected across multiple regions as defined by the United Nations geoscheme (Fig. 1A, C).

From 1996-2011, $bla_{IMP-1}$, $bla_{IMP-4}$ and $bla_{IMP-6}$ emerged in Asia and Australia, together accounting for 79.44% genomes studied. $bla_{IMP-1}$ was identified across Eastern and South Eastern Asia (939/1155 $bla_{IMP-1}$ genomes) during the entire study period, most predominantly Japan, Singapore and China. After 2014, regional $bla_{IMP-1}$ outbreaks were increasingly noted outside of Asia, including in Europe, Western Africa and North America. $bla_{IMP-4}$ was initially noted in China in 1998 and continued to be isolated there through the study period (379/1592 $bla_{IMP-4}$ genomes). However, from 2002, it was well established in Australia (1105/1592 genomes), predominantly on the east coast. Similarly to $bla_{IMP-1}$, sporadic $bla_{IMP-4}$ outbreaks were noted outside these regions between 2014–2023 in Europe and Northern America (62/1592 $bla_{IMP-4}$ genomes). $bla_{IMP-6}$ was identified in Japan from 2000 and remained focused there until end of the study period (224/235

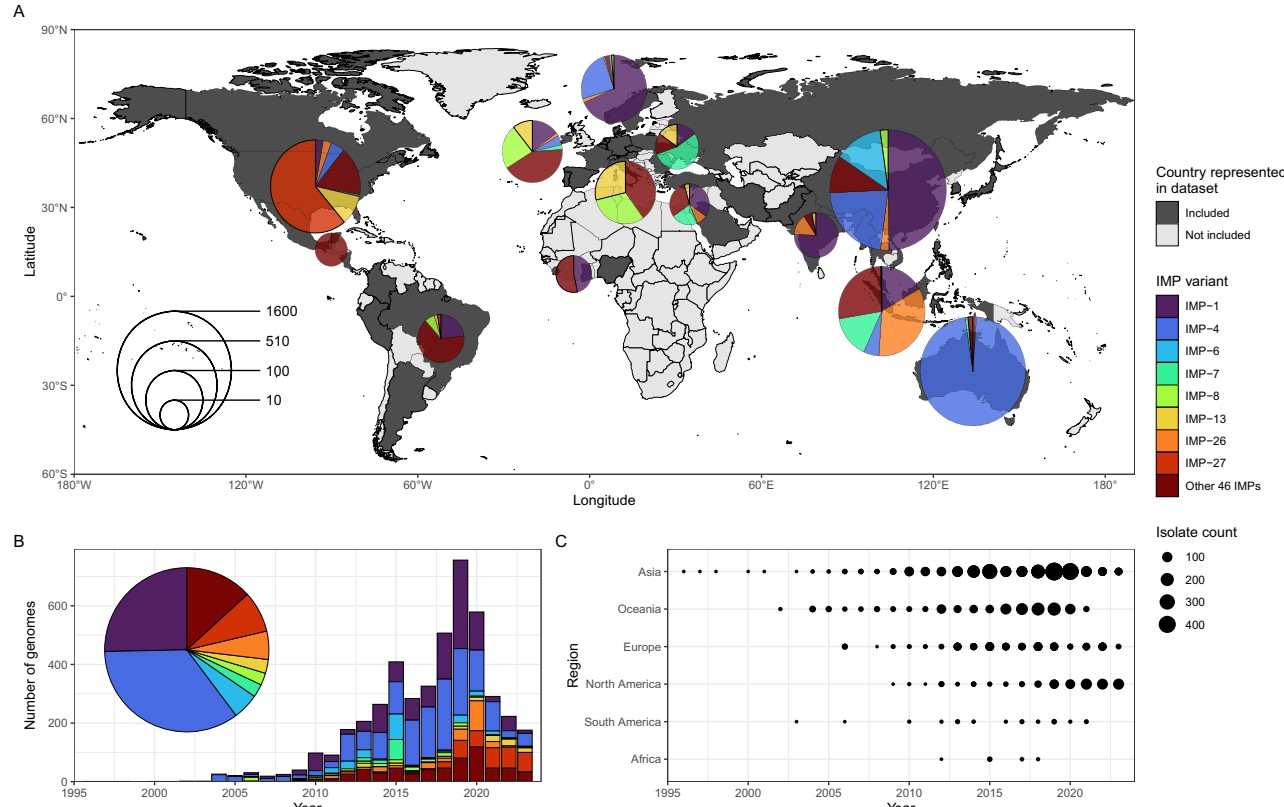

**Fig. 1 | Geographical and temporal spread of $bla_{IMP}$ carbapenemase genes. A** World map showing breakdown of $bla_{IMP}$-carrying genomes and their global species distribution. Size of pie charts indicates number of isolates. $bla_{IMP}$ variants with <100 total observations are grouped into 'Other 46 IMPs'. **B** Global prevalence of most common $bla_{IMP}$ variants over time. **C** Dot plot showing prevalence of $bla_{IMP}$-positive genomes within the United Nations geoscheme regions over time. Source data are provided as a Source Data file.

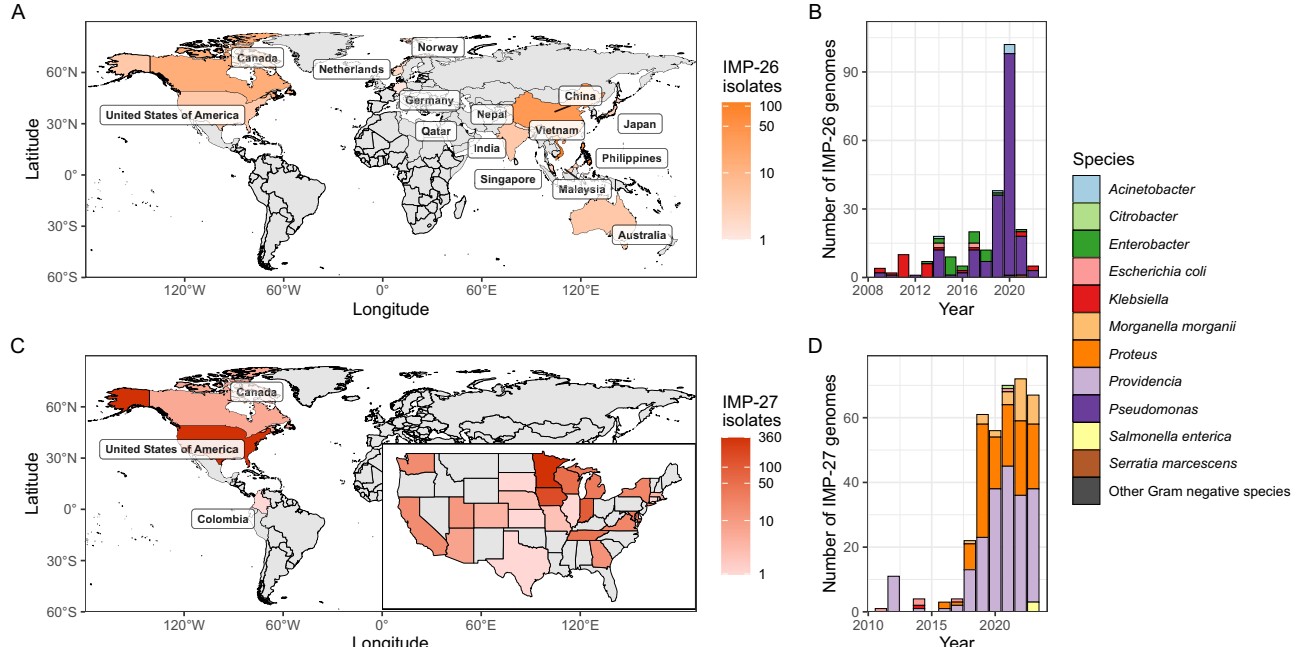

**Fig. 2 | *bla*IMP-26 and *bla*IMP-27 epidemiology. A** Global distribution of *bla*IMP-26, showing regional endemicity in Asia and putative travel-association with other geographic regions. **B** Species breakdown of *bla*IMP-26 genomes over time. **C** Global distribution of *bla*IMP-27, with panel showing regional endemicity concentrated within the US. **D** Species breakdown of *bla*IMP-27 genomes over time. Source data are provided as a Source Data file.

*bla*IMP-6 genomes), with sporadic isolation in South Korea, UK and USA between 2017–2023 (*n* = 11 genomes).

From 2009, *bla*IMP-26 and *bla*IMP-27 emerged as regional *bla*IMP genes in South Eastern Asia and North America, respectively (Fig. 2). *bla*IMP-26 was first noted in 2009 and became established in *P. aeruginosa* in Vietnam and Philippines. *bla*IMP-26 was highly associated with *P. aeruginosa* ST235 (151/254 *bla*IMP-26 genomes). This variant-lineage combination led to subsequent spread internationally across six geographic subregions from South-Eastern Asia to Southern Asia, Australia, Northern Europe, Western Europe and Northern America from 2009-2022, possibly due to travel-associated importation (Fig. 2A, Supplementary Data 3). *bla*IMP-27 emerged from 2011 as the dominant carbapenemase in North America, specifically the US (genomes from 21 states) (Fig. 2B). In addition to geographical location, this *bla*IMP variant displayed a distinct epidemiology characterised by associations with agricultural animals from 2011-2014 before emerging in human-origin genomes from 2016 onwards (Supplementary Data 3). 77.6% (287/370 North American *bla*IMP-27 genomes resulted from expansion of local lineages of *Providencia rettgeri* spp. 1, 2 and 3, *Providencia stuartii* spp. 1 and 2, *Providencia huaxiensis*, *Proteus mirabilis* and *Morganella morganii*.

From 2018 to 2023, several *bla*IMP variants achieved global (*bla*IMP-1, *bla*IMP-4, *bla*IMP-7, *bla*IMP-8, *bla*IMP-13) and regional (*bla*IMP-6, *bla*IMP-26, *bla*IMP-27) endemicity, whereby there was evidence of ongoing spread outside of sporadic outbreaks (Supplementary Data 4). During this period, the global dissemination of *bla*IMP carbapenemase genes was demonstrated with their detection in 42 countries spanning all regions, with 23 countries reporting ≥2 *bla*IMP variants and 8 countries reporting ≥5 *bla*IMP variants.

### *bla*IMP carbapenemase genes found in diverse bacterial hosts with over-representation of multidrug-resistant lineages

Having determined that *bla*IMP genes achieved global spread, we wanted to understand how genomic factors shaped this spread and hence adopted a multi-level approach focusing on bacterial hosts, plasmids and finally other mobile genetic elements. On a bacterial host level, there were 93 species carrying 4556 total *bla*IMP genes but we noted that *bla*IMP variants were associated with specific species (Fig. 3A, Supplementary Data 5). *bla*IMP-1, *bla*IMP-4 and *bla*IMP-6 were predominantly associated with *Enterobacterales* (specifically *E. hormaechei, K. pneumoniae, E. coli*), while *bla*IMP-7, *bla*IMP-13 and *bla*IMP-26 were predominantly noted in *P. aeruginosa*. *bla*IMP-27 had a unique epidemiology dominated by *Providencia, Proteus* and *Morganella* spp.

*P. aeruginosa* sequence type (ST) 235 was the most frequent single lineage isolated (391/4556 genomes, 8.6%) and carried the greatest number of *bla*IMP variants (*n* = 17) (Supplementary Data 5). *P. aeruginosa* ST235 is a global multidrug-resistant (MDR) lineage, recognised for its ability to harbour a high diversity of acquired resistance genes[24]. While there was a close association with *bla*IMP-26 (151/391 *P. aeruginosa* ST235 genomes), ST235 also carried other key *bla*IMP variants including *bla*IMP-1 (55/391 genomes), *bla*IMP-51 (51/391 genomes), and *bla*IMP-7 (46/391 genomes). We noted *bla*IMP presence in several other MDR lineages: *E. hormaechei* ST78 had a close association with *bla*IMP-1 (162/4556 [3.6%] genomes, 160 from Japan); *E. coli* ST131 with *bla*IMP-6 (64/4556 [1.4%] genomes, all from Japan); *K. pneumoniae* ST307 harboured *bla*IMP-38 in a limited number of genomes (*n* = 22, 21/22 from China) (Fig. 3B, Supplementary Data 6).

We then quantified the impact of clonal bias and potential outbreaks by only including a single representative from each cluster of closely-related genomes (i.e. 'dereplication') (see 'Methods'). We defined these 'IMP-clusters' as clusters of genomes which shared the same *bla*IMP variant, species, lineage (ST or clonal group [CG], if no MLST schema available) and were within a species-specific threshold of 5 single-nucleotide variants (SNV) per Mb (see 'Methods'). This resulted in 1381 IMP-clusters, with 3175/4556 (69.9%) closely-related genomes removed by dereplication (Fig. 3B, Supplementary Data 6). Indeed, the majority of *bla*IMP genomes were clonally linked to at least one other genome: 700/1,381 IMP-clusters contained >1 genome (Fig. 3B, Figure. S1), potentially reflecting sequencing conducted in outbreak settings. This included multiple *P. aeruginosa* ST235 IMP-clusters (*bla*IMP-26 – 127 genomes, *bla*IMP-51 – 51 genomes, *bla*IMP-1 – 50 genomes, *bla*IMP-31 – 39 genomes) (Figure. S2A, Supplementary Data 6). Some IMP-clusters also accounted for high proportions of observations for that gene: *K. pneumoniae* ST37 and *E. coli* ST131 *bla*IMP-6 IMP-

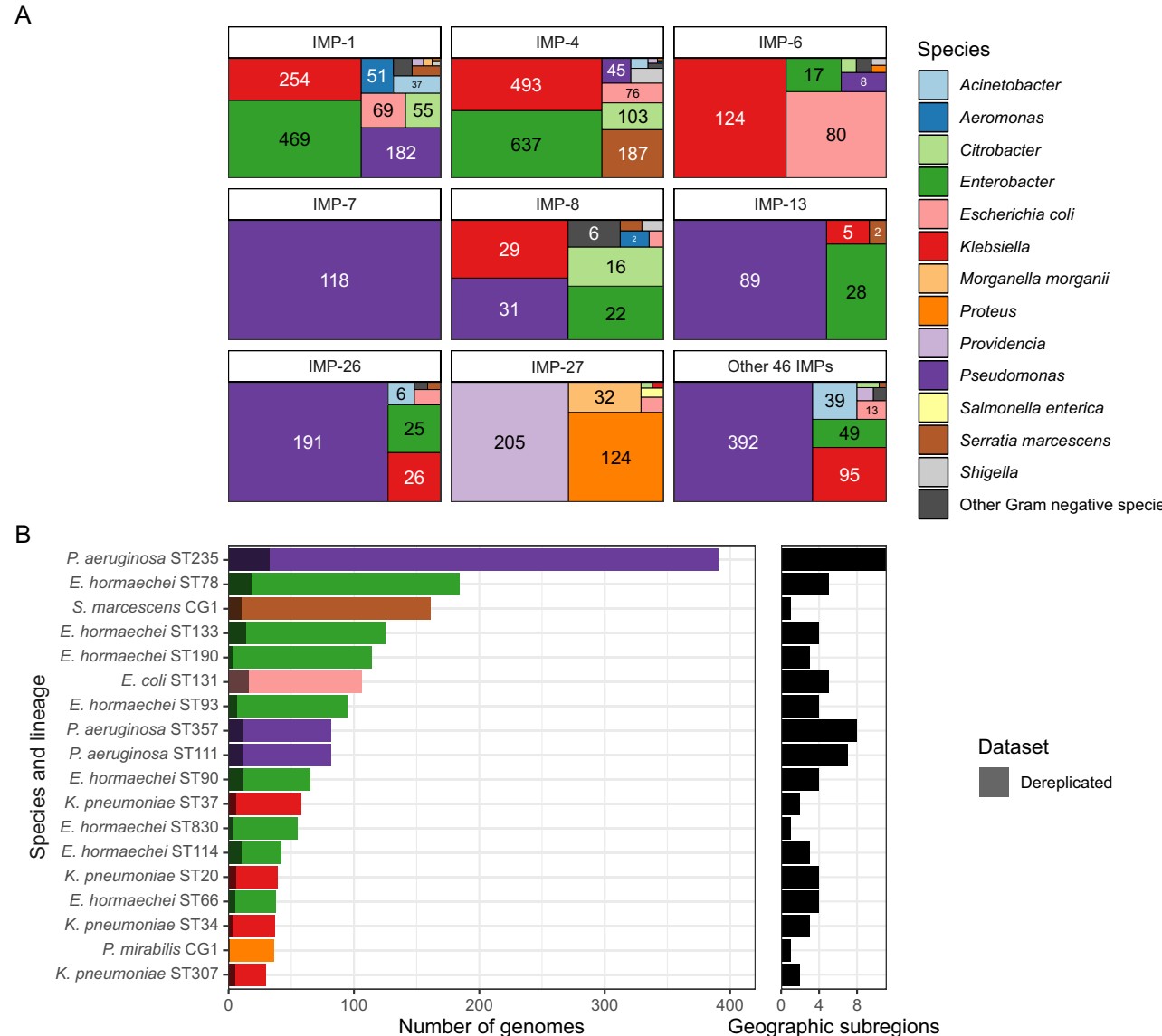

**Fig. 3 | Species and lineages associated with *bla*<sub>IMP</sub> carriage. A** Species breakdown across each *bla*<sub>IMP</sub> variant. **B** Highly prevalent lineages within the dataset, showing the impact of dereplicating IMP-clusters. Dereplicated bars are shown as grey overlays over coloured genome counts, not stacked bars. Only lineages with ≥30 genomes are shown. Number of geographic subregions each lineage was detected in is shown as a companion plot. Full data found in Supplementary Data 5 and Supplementary Data 6.

clusters accounted for 14.5% and 13.2% of all *bla*<sub>IMP-6</sub> observations, while *E. hormaechei* ST190 and ST78 IMP-clusters accounted for 6.91% and 6.67% of *bla*<sub>IMP-4</sub> and *bla*<sub>IMP-1</sub> observations. We then assessed IMP-clusters for possible spread between countries and geographic subregions and noted that only 108/1381 (7.8%) were found across >1 country. From the perspective of regional spread, 5.7% (78/1381) IMP-clusters were found across ≥2 geographic regions, including 24 independent *P. aeruginosa* IMP-clusters (Supplementary Data 6).

### Diverse plasmids facilitate global dissemination of *bla*<sub>IMP</sub> carbapenemase genes

To determine the genetic context of *bla*<sub>IMP</sub> genes, we first analysed all available long-read genomes in the dataset (*n* = 536) (see 'Methods'). *bla*<sub>IMP</sub> genes were located on plasmids in 433/536 genomes (81.3%). These were divided into 52 plasmid 'clusters' as defined by MOB-suite plasmid replicon clusters[25], which carried a total of 20 *bla*<sub>IMP</sub> variants (Supplementary Data 7). IncHI2A, IncC and IncN plasmids collectively accounted for 56.0% (244/433) plasmids and each carried four, two

and five different *bla*<sub>IMP</sub> variants, respectively. These plasmids were found across Asia and Australia with sporadic cases in Europe (Table 1 and Supplementary Data 7). Other key *bla*<sub>IMP</sub> plasmids were IncU, IncFIB, IncFIB/IncFII-type 2, IncL/M, IncP and an untyped *A. baumannii* plasmid. These carried 2–4 *bla*<sub>IMP</sub> variants and were found in ≥2 geographic regions. Three long-read genomes carried two IMP variants, each co-located on the same molecule (Unknown rep plasmid, chromosomally, and IncFIA:IncFIB, respectively) (Supplementary Data 1).

There were 181 plasmid cluster-*bla*<sub>IMP</sub> variant combinations and we evaluated their geographic distributions. We noted diversity of IncN, IncHI2A and IncP plasmids, with multiple distinct plasmid clusters (*n* = 8, *n* = 5 and *n* = 4, respectively). These were predominantly associated with one *bla*<sub>IMP</sub> variant and one region (Supplementary Data 7). Only 54/181 *bla*<sub>IMP</sub> variant-plasmid cluster combinations were found across different countries, while 13/181 were found in ≥3 geographic regions, driven by successful expansion of key IMP-clusters carrying broad host-range plasmids (IncHI2A, IncL/M, IncC, IncN). This indicated that while subregional spread of closely-related plasmids

**Table 1 | Summary of key plasmid clusters (n ≥ 100) and associated features. Full data can be found in Supplementary Data 1 and Supplementary Data 7**

| MOB-suite cluster | Inc replicon | Year range | $bla_{IMP}$ variant | Countries | Species |
|---|---|---|---|---|---|
| AA739_AJO55 | IncHI2A (n = 396) | 2004–2022 | IMP-4, IMP-26, IMP-1, IMP-22, IMP-13 | Australia, Philippines, Japan, Ireland, UK, Spain, US | K. michiganensis, K. oxytoca, E. hormaechei, K. pneumoniae, E. cloacae, C. youngae, E. asburiae, E. kobei, R. planticola, E. roggenkampii, E. coli, C. amalonaticus, K. aerogenes, C. freundii, E. bugandensis, K. variicola, S. boydii, S. sonnei, C. koseri, C. farmeri, E. chengduensis, C. portucalensis |
| AA552_AJ753 | IncN (n = 302) | 2004–2023 | IMP-6, IMP-1, IMP-66, IMP-26, IMP-22, IMP-4, IMP-74, IMP-11 | Japan, Philippines, UK, Singapore, Spain, Portugal, Germany, Peru, China | K. pneumoniae, E. coli, C. freundii, E. asburiae, E. hormaechei, E. kobei, E. ludwigii, K. michiganensis, R. ornithinolytica, K. variicola, S. sonnei, E. chengduensis, P. stuartii Spp 1, L. adecarboxylata, P. mirabilis, K. quasipneumoniae, E. cloacae, K. grimontii, C. braakii |
| AA739_AJO57 | IncHI2A (n = 276) | 2007–2021 | IMP-1, IMP-13 | Japan, US, UK | E. hormaechei, C. freundii, E. asburiae, E. kobei, K. michiganensis, E. cloacae, E. coli, E. chengduensis, K. pneumoniae, S. marcescens, E. roggenkampii, K. oxytoca |
| AA860_AJ266 | IncC (n = 260) | 2002–2022 | IMP-4, IMP-1, IMP-60, IMP-8, IMP-15, IMP-23 | Australia, Japan, Netherlands, UK, Spain, China, Brazil, Singapore | S. marcescens, K. pneumoniae, E. chengduensis, E. hormaechei, K. oxytoca, C. freundii, E. asburiae, E. coli, V. alginolyticus, K. michiganensis, K. variicola, S. sonnei, E. cloacae, P. mirabilis, C. koseri, E. bugandensis, K. quasipneumoniae |
| AAOO2_AH532 | IncL/M (n = 253) | 2006–2023 | IMP-4, IMP-34, IMP-1, IMP-22, IMP-59 | Australia, Japan, Philippines, Portugal, US, UK | K. pasteurii, K. quasipneumoniae, C. freundii, K. pneumoniae, E. hormaechei, E. coli, K. michiganensis, S. sonnei, S. marcescens, C. murliniae, K. aerogenes, E. asburiae, K. variicola, S. boydii, C. koseri, E. kobei, L. adecarboxylata, A. subterranea, C. farmeri |
| AA552_AJ757 | IncN (n = 153) | 2008–2023 | IMP-6, IMP-4, IMP-38, IMP-26 | Japan, China, US | K. pneumoniae, K. michiganensis, K. pasteurii, A. subterranea, C. freundii, E. asburiae, E. hormaechei, K. grimontii, E. coli, E. soli, R. ornithinolytica, K. quasipneumoniae, K. variicola, R. planticola, C. amalonaticus, C. portucalensis, H. chinensis, C. braakii, E. roggenkampii |
| AA739_AJO59 | IncHI2A (n = 134) | 2008–2023 | IMP-6, IMP-1, IMP-4, IMP-22, IMP-8, IMP-13, IMP-26, IMP-19 | UK, Portugal, Taiwan, US, China, Australia, Poland | C. youngae, E. hormaechei, E. asburiae, A. hermannii, E. cloacae, E. coli, K. pneumoniae, E. bugandensis, K. oxytoca, C. freundii, E. kobei, K. aerogenes, S. marcescens, P. vulneris |
| AA739_AJO58 | IncHI2A (n = 113) | 2010–2022 | IMP-4, IMP-6, IMP-1, IMP-13 | Australia, Japan, Spain, US, Singapore, China | E. hormaechei, K. oxytoca, K. pneumoniae, E. coli, E. asburiae, K. michiganensis, E. kobei, C. amalonaticus, K. variicola |

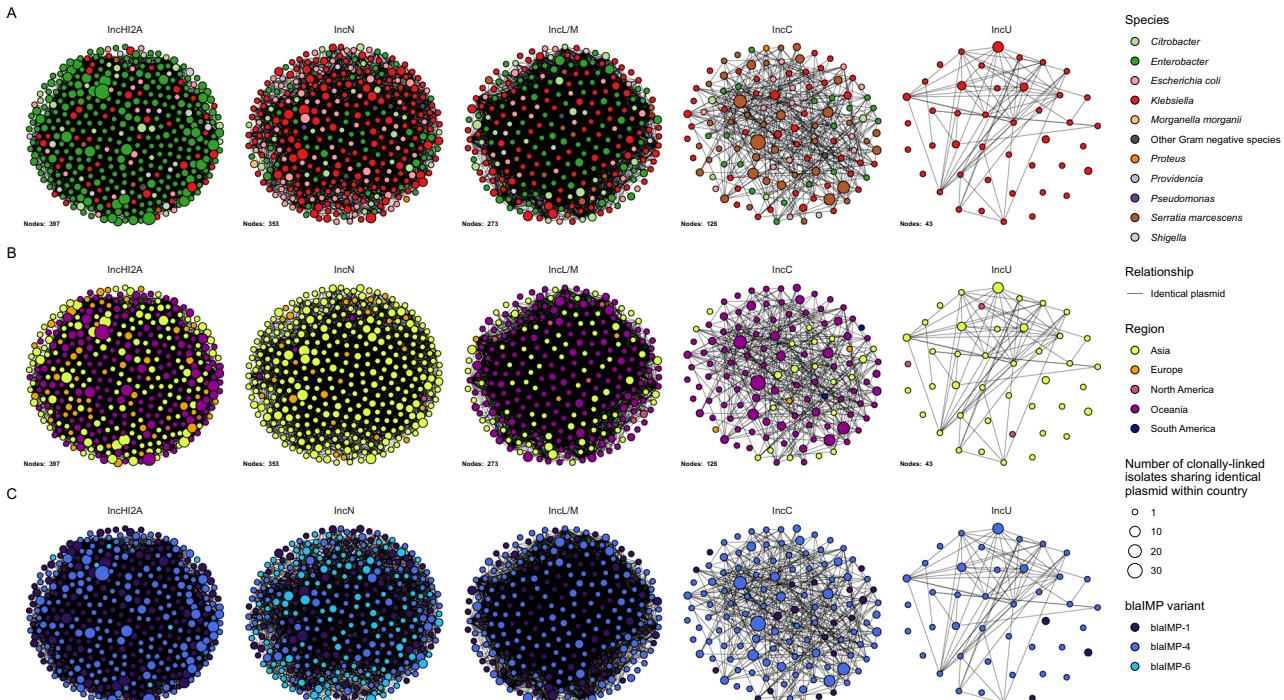

**Fig. 4 | Key *bla*~IMP~ plasmids by bacterial species and geographic region.**
**A** Network of related plasmids, separated by plasmid Inc types and coloured by host species. Edges show identical plasmids between IMP-clusters (nodes). Size of node represents number of genomes within an IMP-cluster and country also carrying the same plasmid. **B** Identical networks coloured by geographical region. **C** Identical networks coloured by *bla*~IMP~ variant.

may have occurred, spread between regions was not detected in most cases. The regional *bla*~IMP-26~ gene was found in IncHI2A, IncU and five untypeable plasmids (MOB-suite clusters AA739_AJ059, AC212_AL309, AC213_AL312. AC935_AM305 and AD068_AM495), respectively, however long read data was limited (13 genomes). Plasmids also displayed clear, species-specific host ranges. Most notably, IncHI2A plasmids were overwhelmingly associated with *E. hormaechei* (*n* = 101), while IncC plasmids were associated with *S. marcescens* (*n* = 30) and *K. pneumoniae* (*n* = 18).

We then assessed co-occurrence of these 433 *bla*~IMP~-plasmids and non-*bla*~IMP~ plasmids within the same genomes via network analysis. To account for clonal bias, single representatives from each IMP-cluster were used (Figure. S3). Analysis of node degrees (number of edges for each node) revealed that *bla*~IMP-4~ plasmids had the greatest number of non-*bla*~IMP~ plasmid co-associations within genomes. In particular *bla*~IMP-4~ IncC, *bla*~IMP-4~ IncL/M and *bla*~IMP-4~ IncN plasmids had the highest network degrees (37, 33 and 25, respectively). Plasmid co-occurrence and dissemination may be shaped by both bacterial host traits (*E. hormaechei*, *E. coli* and *K. pneumoniae*) and plasmid-specific factors. *Enterobacterales* are known for their ability to carry multiple plasmids[26] and were noted to have high centrality in our data, suggesting that host factors play the defining role in this co-occurrence analysis.

To gain insights from genomes with short-read data only, we clustered short-read contigs to long-read plasmid clusters (see 'Methods'), with the caveat of reduced predictive confidence (Fig. 4). Across the entire short and long-read dataset, *bla*~IMP~ variants were found on plasmids in 2909/4556 (63.8%) genomes, the chromosome in 97 genomes, with 1553 remaining unclassified (Supplementary Data 1). Representation of most plasmids was similar following inclusion of short-read assemblies (Figure. S4). Notable exceptions included IncC plasmids decreasing from 16.3% to 8.3%, while IncN and IncL/M plasmids increased (10.8% to 19.7% and 8.3% to 14%, respectively). IncN increases resulted from short-read *K. pneumoniae* species complex genomes largely absent in the long-read/hybrid dataset, while the IncL/

M increase was driven by additional *Enterobacter* spp., *K. pneumoniae* species complex and *E. coli* short-read genomes. Of the 181 *bla*~IMP~ variant-plasmid cluster combinations, 55 were found across multiple countries. A key example were *bla*~IMP-4~ IncL/M plasmids that were detected in 249 genomes across Australia, US, UK and Philippines in 19 bacterial species (predominantly *Enterobacterales*). The remaining 126/181 *bla*~IMP~ variant-plasmid cluster combinations were specific to a single country.

We then assessed relationships of *bla*~IMP~ plasmids with bacterial host lineages. In our prior work, we noted successful lineage-plasmid pairings that we termed 'propagators'[19]. There were 35 lineages possibly acting as propagators with ≥10 genomes including *S. marcescens* CG1 carrying *bla*~IMP-4~ IncC plasmids (145 genomes), and *bla*~IMP-1~ IncHI2A plasmids associated with *E. hormaechei* ST78 and *E. hormaechei* ST133 (*n* = 116 and *n* = 88, respectively) (Supplementary Data 8). Additionally, we identified 64 'connector' lineages capable of harbouring ≥3 *bla*~IMP~ plasmid clusters (Supplementary Data 8), which could serve as an opportunity for transfer of *bla*~IMP~-containing integrons[19]. *E. hormaechei* ST78 and *E. coli* ST131 were the most prominent, carrying 7 plasmid clusters each (Supplementary Data 8).

*P. aeruginosa* was notable for having the greatest number of chromosomally located *bla*~IMP~ genes of all species (*n* = 59). Of the 977 genomes, 199 harboured plasmid-borne *bla*~IMP~, and 719 were 'unclassified' due to lack of replicon typing. We did not speculate on the unclassified group. Nonetheless, *bla*~IMP~ variants showed strong lineage coupling, such as *bla*~IMP-26~ in ST235, where 136 genomes carried it on unclassified molecules and 22 on plasmids (20/22 with unknown replicon types).

**Mobile elements associated with *bla*~IMP~ carbapenemase genes**
*bla*~IMP~ genes are frequently associated with integrons[27], integrative and conjugative elements (ICEs)[28], insertion sequences (IS)[29] and transposons[30]. We therefore sought to determine if these mobile elements could provide insight into the spread of *bla*~IMP~ genes. We analysed the genetic context up to 10 kb up- and down-stream of each

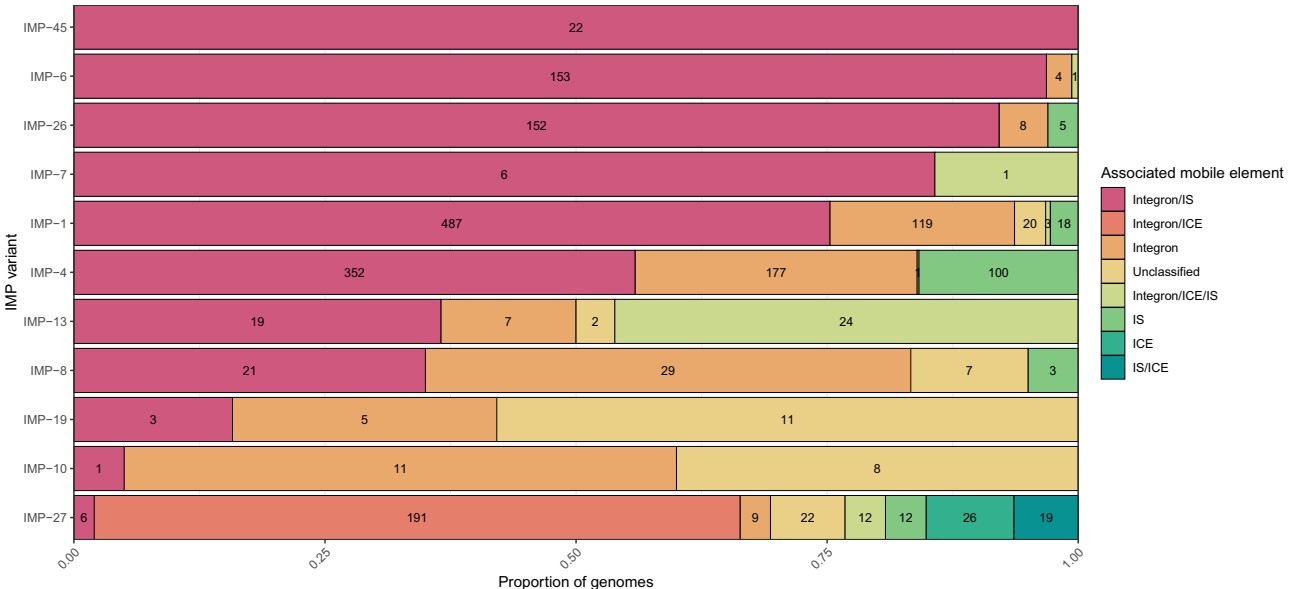

**Fig. 5 | The majority of $bla_{IMP}$ variants are associated with integrons and IS elements.** Column graph showing the proportion of each $bla_{IMP}$ variant and its association with mobile elements including integrative conjugative elements (ICE), integrons, insertion sequences (IS), and unclassified. Numbers within bars show $n$ of each group. Full data found in Supplementary Data 1.

$bla_{IMP}$ gene. To maximise identification of mobile elements and analysis dataset, only contigs ≥6 kb in length were considered for this analysis which resulted in 2314/4559 eligible $bla_{IMP}$-containing contigs (Fig. 5). A threshold of ≥6 kb was used as this was a natural cutoff point when analysing the distribution of $bla_{IMP}$-containing contig lengths (data not shown). Initially, we examined the total presence of mobile elements across the dataset. Of these, 2053/2314 (88.7%) contigs showed direct association with integron elements (850 intact and 1203 with *attC* clusters only) and almost all were class 1 integrons (795/850). When taking other mobile elements into account, integrons were found alone in 478/2314 (20.6%) contigs but were co-located with other mobile elements in the majority, including IS (1332/2314 [57.6%]) and ICE (192/2314 [8.3%]) (Fig. 5). This analysis was limited by the lack of eligible contigs due to the dataset being comprised mostly of short-read assemblies. Contig breaks are commonly caused by the presence of IS and repetitive elements[31], potentially biasing $bla_{IMP}$ flanking region analyses. However, we consistently found intact IS within these flanking regions, showing the validity of this approach.

Co-location of multiple mobile elements with $bla_{IMP}$ genes may have contributed to their dissemination across populations due to increased mobility. Integron/IS-associated $bla_{IMP}$ genes were most frequent (across 56 species, 39 IMP variants and 41 plasmid clusters), dating back to the earliest $bla_{IMP}$ isolates in the late 1990s (Figure. S5). We saw no major differences between $bla_{IMP}$ variants and their putative associated mobile element/s. $bla_{IMP-27}$ was a notable exception: it had the most variable mobile element associations, characterised by presence of ICE alone or in combination with integrons and/or other IS (Fig. 5).

**IMP variants display convergent evolutionary trends**
Beyond analysing genomic contributors to IMP dissemination, we analysed sequence and structural variation of IMP variants in our study (Fig. 6A and B). Critically, this variation is closely linked with β-lactam hydrolytic specificity and alters minimum inhibitory concentrations of carbapenems[32–35]. This led us to hypothesise that it may influence $bla_{IMP}$ dissemination. Predicted structures for each mature IMP variant were found have high confidence, with a mean predicted local distance difference test (plDDT) of 96.36 ± 7.23 SD across all residues and variants (Supplementary Data 9). When comparing IMP variant structures

to each other, we found they were highly structurally conserved, with a mean lDDT score of 0.977 (Fig. 6C). Despite this, we saw reduced amino acid conservation scores at key positions throughout the structure (Fig. 6D), indicative of varying substrate specificity/activity. We examined mutations in key residue positions[30,31,36,37],(150,167,196) previously shown to impact carbapenem hydrolytic specificity and noted a consistent pattern of convergent evolutionary mutations throughout the protein phylogeny at these key residues. Many of these convergent mutations appear to have been acquired independently (Fig. 6A), such as 31 F found in multiple IMP sequences.

**Non-human reservoirs of $bla_{IMP}$ carbapenemase genes**
We then adopted a One Health approach by analysing sources of $bla_{IMP}$ genomes to determine the potential contribution of environmental or animal reservoirs to $bla_{IMP}$ dissemination. Genomes of human origin accounted for the majority (4051/4556, 88.9%), while genomes of environmental and animal origin accounted for 10.0% (454/4556) and 1.1% (51/4556), respectively (Fig. 7A). Nine samples were unable to be classified. Most environmental genomes (353/454, 77.8%) came from healthcare settings, with at least 128/353 (36.3%) coming from hospital aquatic environments (Fig. 7B, Supplementary Data 10). Animal data were limited but $bla_{IMP-4}$ and $bla_{IMP-38}$ were detected in Australian seagulls ($n = 31$ and $n = 1$, respectively)[38], and $bla_{IMP-27}$ in US and Canadian genomes of pig origin ($n = 4$ and $n = 1$, respectively). 65/1,381 (4.7%) IMP-clusters were found across multiple source categories (Fig. 7A), most notably $bla_{IMP-26}$ *P. aeruginosa* ST235 and $bla_{IMP-4}$ *S. marcescens*. Only three IMP-clusters had genomes of both human and animal origin (all birds), including $bla_{IMP-4}$ *E. coli* ST58, $bla_{IMP-27}$ *P. rettgeri* spp. 2 CG5 and $bla_{IMP-45}$ *P. aeruginosa* ST313. We then examined the transmission of plasmids between sources, after accounting for IMP-clusters. We identified at least 20 independent plasmids moving between sources, most commonly between clinical and environmental categories (15/20). These included $bla_{IMP-4}$ ($n = 15$), $bla_{IMP-6}$ ($n = 2$), $bla_{IMP-1}$ ($n = 1$) and $bla_{IMP-8}$ ($n = 1$) (Supplementary Data 11). $bla_{IMP-4}$ IncN plasmids were the most notable, spreading between clinical and environmental source categories via 36 individual clones across eight species. We also found one case of a $bla_{IMP-4}$ IncHI2A plasmid moving between clinical, environmental and animal source categories across 13 genomes from six species.

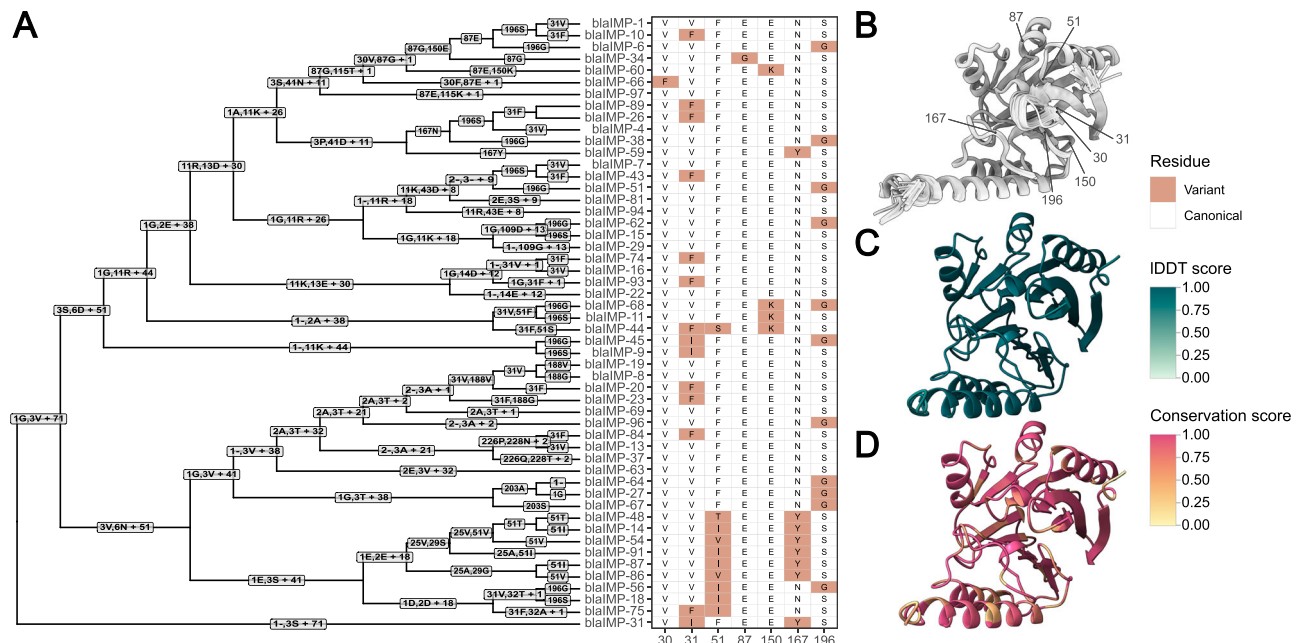

**Fig. 6 | IMP variants have key residue changes which alter catalytic activity.**
**A** Multiple sequence alignment guide tree with associated amino acid changes. Key residues which have demonstrated impact on antimicrobial resistance profiles[32,35,51,52] are shown as a heatmap. **B** Predicted AlphaFold2 structures of each 52 IMP variants, structurally aligned. **C** lDDT scores for each residue after structural alignment with FoldMason, indicating level of structural conservation. **D** Amino acid conservation score, as calculated using residue similarity and the bio3d sub matrix. The Multiple Sequence Alignment is found at Figshare: https://doi.org/10.6084/m9.figshare.28440992.

## Discussion

We have systematically determined the global distribution, diversity and epidemiology of 52 $bla_{IMP}$ variants across 4,556 publicly-available genomes. This enabled us to reconstruct a complete picture of the emergence of IMP carbapenemases as a multi-faceted problem across all geographic regions. Previous studies have either focused on a single $bla_{IMP}$ variant or geographical site[5,19,22,39–41], leaving gaps in our understanding of the genomic epidemiology of these genes. Our analysis has revealed the under-recognised diversity and distribution of $bla_{IMP}$ carbapenemase genes. We have shown that $bla_{IMP-1}$ and $bla_{IMP-4}$ were the most common variants due to their early emergence[42,43] and wide dissemination, achieving global endemicity along with $bla_{IMP-7}$, $bla_{IMP-8}$, $bla_{IMP-13}$ (Fig. 1). In contrast, several $bla_{IMP}$ variants displayed distinct regional patterns including $bla_{IMP-26}$ (Fig. 2A) in Southeast Asia and $bla_{IMP-27}$ in North America (Fig. 2C), achieving regional endemicity. These patterns have been previously unrecognised and challenge earlier perceptions that IMP carbapenemases are largely confined to the Asia-Pacific region[44].

Horizontal gene transfer, more than clonal spread, enabled global dissemination of $bla_{IMP}$ genes. While previous work has demonstrated associations of $bla_{IMP}$ and $bla_{GES-5}$ genes with integrons[19,45,46] and co-location with IS[47,48], we demonstrate these intersections systematically at scale. We found that $bla_{IMP}$ genes were almost invariably embedded in class 1 integrons, frequently flanked by IS. This genetic context promoted $bla_{IMP}$ mobility, enabling entry into a vast array of at least 52 plasmid clusters. These included broad host-range IncHI2A, IncC, and IncN plasmids, all found to carry multiple $bla_{IMP}$ variants. The coupling of integrons and IS with these broad-host range plasmids likely drove interspecies spread of $bla_{IMP}$ genes into diverse bacterial hosts, specifically Enterobacterales (e.g. Enterobacter spp., Klebsiella spp. but also Providencia spp. and Proteus spp.). The regional $bla_{IMP-27}$ gene was not associated with any known plasmids using our approach (which relied on completed plasmid references), but has been shown to be associated with the conjugative pPM187 (IncX8 replicon type) and pPR1 (no

known replicon type) plasmids, allowing experimentally-verified inter-species spread[49].

Despite horizontal gene transfer being a key driver of $bla_{IMP}$ spread, we also noted that proliferation of certain bacterial host lineages was an important contributor to $bla_{IMP}$ dissemination. $bla_{IMP-1}$ P. aeruginosa ST235 was a successful global IMP lineage but other lineages were more geographically limited. Instead, we noted numerous local outbreaks and limited inter-country spread of successful plasmid-lineage combinations that we previously termed 'propagators'[19]. We identified at least 35 propagators that sustained and amplified local $bla_{IMP}$ outbreaks (e.g. $bla_{IMP-4}$ S. marcescens-IncC, $bla_{IMP-1}$ E. hormaechei-IncHI2A). This is in contrast to other carbapenemases such as KPC and OXA-48, which are associated with globally disseminated clones such as K. pneumoniae ST258 and ST11, respectively[50]. In addition, we propose the idea of 'connectors': lineages that can accept multiple plasmids and serve as bridges for $bla_{IMP}$ gene transfer through mobile genetic elements, without themselves causing local outbreaks. We identified 64 such lineages such as E. hormaechei ST78 and E. coli ST131. The interplay of propagators and connectors helps explain how $bla_{IMP}$ genes can disseminate and also repeatedly establish in new hosts. The chromosomal integration of $bla_{IMP}$ in P. aeruginosa likely facilitates stable inheritance and persistence within successful clones, supporting ongoing clonal spread. This is consistent with reports of chromosomal integration of other key resistance genes, such as $bla_{CTX-M-15}$, which similarly enhances stability and long-term maintenance within bacterial populations[36]. The $bla_{IMP}$ threat is therefore multidimensional: it spreads by many local expansions rather than a single dominant lineage crossing borders, thus posing difficulties for both detection and control.

Beyond $bla_{IMP}$ gene transfer, we also analysed the structures of IMP enzymes to examine whether adaptive changes themselves may be contributing to spread. Despite the diversity of 52 known IMP variants, we detected convergent evolutionary patterns, with repeated missense mutations at specific sites. While not every residue across

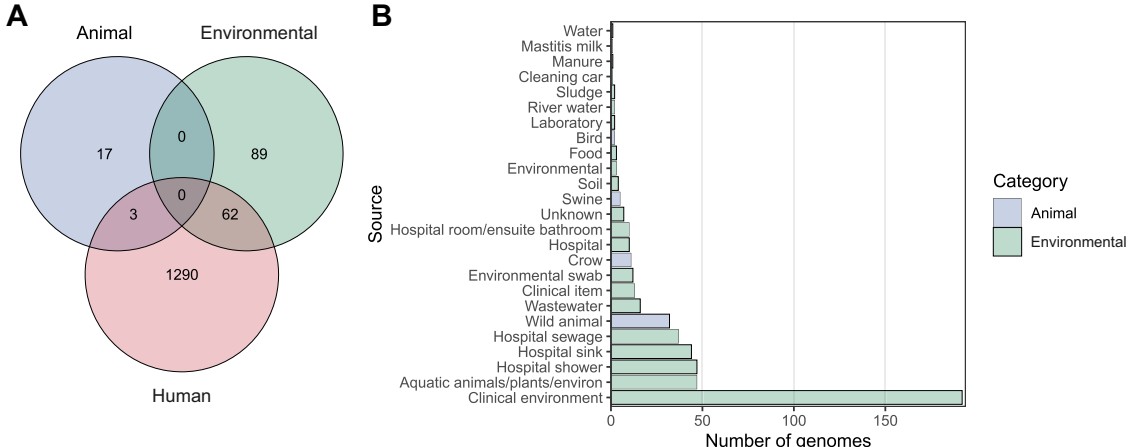

**Fig. 7 | Human, animal and environmental origins of $bla_{IMP}$ genomes. A** Venn diagram showing the intersections of IMP-clusters (as previously defined) between genomes of human, environmental and animal origins, indicating either clonal spread between categories or discrete isolation. **B** Number of genomes isolated from specific non-human sources. Source data are provided as a Source Data file.

each variant has been studied within the context of AMR, these changes likely offer functional advantages by altering carbapenem hydrolytic activity and thus minimum inhibitory concentrations, as previously seen for V31F, S196G and N167Y[32,33,35,51–53]. Many of these convergent mutations appear to have been acquired independently, such as 31F found in divergent $bla_{IMP-10}$ and $bla_{IMP-26}$ sequences (Fig. 6) indicating a common evolutionary solution that allows the successful proliferation of IMP variants. We identified IMP-26, as a key regional endemic clone in Asia, which is more effective at hydrolysing meropenem and doripenem than older variants and thus displays a broader and more effective carbapenemase phenotype[54]. It is reasonable to speculate that $bla_{IMP-26}$ evolved from $bla_{IMP-4}$, given their single nucleotide polymorphism difference while sharing similar genetic contexts[55]. Indeed, our dataset shows $bla_{IMP-4}$ was first noted genomically in 1998, followed by $bla_{IMP-26}$ in 2009. While $bla_{IMP-4}$ outnumbered $bla_{IMP-26}$ ($n = 1592$ vs $n = 254$), they share overlap of six geographical subregions and 13/14 species carrying $bla_{IMP-26}$ can also carry $bla_{IMP-4}$, raising the possibility that there have been multiple independent $bla_{IMP-26}$-evolution events. While we cannot determine which selective pressures led to the rise of each $bla_{IMP}$ variant, we speculate that this could be driven by antimicrobial usage[32] and species-specific susceptibility[56]. The fact that similar mutations have arisen independently in different $bla_{IMP}$ variants implies ongoing adaptive evolution. Similar convergent mutation patterns have been shown in the *K. pneumoniae* extended-spectrum β-lactamase $bla_{SHV}$[57]. These evolutionary patterns raise concern that IMP enzymes may continue to evolve, not only with increasing carbapenemase activity but also potentially evading even the newest β-lactamase inhibitors.

Finally, we adopted a One Health lens to help understand and hopefully control $bla_{IMP}$ spread. Although most genomes came from human clinical isolates, $bla_{IMP}$ genes were also found in hospital environments (e.g., surfaces, wastewater), as well as animal samples (e.g., in cats and birds). Our fine-grained IMP-cluster analysis allowed detection of movement between these One Health categories, predominantly between human clinical isolates and clinical environments, and to a lesser extent between environmental/animal sources. This illustrates how healthcare-associated resistance genes can spill over into animals or the environment, creating secondary reservoirs. These reservoirs may, in turn, seed new infections back into humans, forming a cycle that blurs traditional epidemiological boundaries. The presence of $bla_{IMP}$ genes in *Serratia* spp., *Pseudomonas* spp. and other environmental bacteria also raises the possibility of environmental persistence. $bla_{IMP}$ genes therefore have access to multiple

ecological niches (human, animal, and environmental) that constantly interact, further complicating control efforts. Furthermore, non-clinical isolation sources are likely under-sampled, thus underestimating the true extent of non-human $bla_{IMP}$ reservoirs and underscoring the need for a One Health approach for surveillance and control.

Our study had ambitious reach, aiming at analysing the global genomic epidemiology of $bla_{IMP}$ genes, but this was also the source of several limitations. Our analysis relied on publicly-available genomes that represented biases in whole genome sequencing activity and genomic surveillance. This led to overrepresentation of countries with genomic infrastructure, with corresponding under-representation of low- and middle-income countries that are likely disproportionately impacted by infections caused by IMP carbapenemases[58]. On a technical level, we were reliant on the quality of genomic data submitted. Most data were derived from short-read sequencing, introducing uncertainty for determining the genetic contexts of $bla_{IMP}$ genes. We tried to compensate for this by conducting a mapping-based analysis but acknowledge the inherent limitations of this approach. Similarly, metadata were frequently poorly curated, thus preventing discovery of additional non-clinical spread and limiting geographic/temporal inferences. Moreover, the dataset includes 52 of 96 known IMP variants, leaving rare variants uncharacterised.

In summary, we have demonstrated that the emergence of IMP carbapenemases has largely 'flown under the radar', despite the establishment of $bla_{IMP}$ variants with global endemicity (IMP-1, IMP-4, IMP-7, IMP-8, IMP-13), as well as regional endemicity (IMP-26 and IMP-27). We have shown that this spread was the result of a complex interplay of genomic drivers of dissemination (at the mobile genetic element, plasmid and bacterial host lineage levels), with horizontal gene transfer playing a more substantial role than specific lineages. We noted convergent evolution in IMP carbapenemase enzymes, suggesting adaptation under evolutionary pressures and posing an ongoing challenge with the possibility of adapting to and conferring resistance to new antimicrobial treatments. Finally, we noted the presence of One Health reservoirs of IMP carbapenemases, although detailed analysis was limited by the small number of non-clinical genomes. These findings provide a detailed atlas of IMP carbapenemases and their global spread that also carry implication for other carbapenemases and emerging forms of antimicrobial resistance. In addition to casting light on the extent IMP carbapenemases have silently become a global problem, we have provided a roadmap for future interventions to disrupt their future spread. Our work highlights the need for more robust and sophisticated surveillance approaches

that address gaps in low- and middle-income countries, incorporate methodology to detect plasmid and mobile genetic element transmission and conduct more systematic sampling of One Health reservoirs. This enhanced surveillance needs to be linked to targeted infection prevention and antimicrobial stewardship activities to reduce IMP cases and transmission. We have carried out a systematic analysis of IMP carbapenemases using three decades of data but the critical threat they pose now requires a proactive, real-time and co-ordinated public health response.

## Methods

### Genome acquisition, species identification and genotyping

NCBI Pathogen Isolate Browser[59] was searched using the term "AMR_genotypes:blaIMP*" to retrieve assemblies containing all IMP variants. This initial search identified a total of 4063 genomes from 517 BioProjects. Of these, 3177 assemblies were downloaded from Pathogen Isolate Browser and 886 reads were obtained from Sequence Read Archive (SRA) and assembled with Unicycler v.0.4.8[31] with standard parameters.

To identify additional publicly available assemblies, protein accession numbers for all IMP variants were obtained and queried against NCBI GenBank databases using BLAST v2.15[60]. Assemblies retrieved from this search were compared against the Pathogen Isolate Browser dataset, and non-duplicates were retained, resulting in an additional 583 assemblies. Collectively, this approach yielded a total of 4,556 assemblies, which were used for all subsequent analyses. Metadata was extracted from the BioSample accessions. In cases where no 'collection date' was available and a BioProject did not contribute ≥50 isolates, the date of sequence upload was used as a proxy. Assembly quality was checked using Quast v5.3.0[61], and all genomes with an N50 ≥ 10,000 and ≤1000 contigs were included (Supplementary Data 1). Species identification was performed using Speciator[62]. Genomes were annotated using Prokka v1.14.6[63]. We then performed resistance gene detection with AMRFinderPlus v3.12.8[64]. IMP variants were only considered if they had a 100% match for both identity and query coverage (Supplementary Data 1). In one case (SRR17656613), a partial match was found, which was confirmed via minimap2 v2.26[65] alignment of $bla_{IMP}$ alleles to contigs linked within the assembly graph. We determined in silico multi-locus sequence type (ST) using 'mlst' v.2.19.0[66]. All inconclusive ST calls were checked with SRST2 v0.2.0[67].

We note that several key $bla_{IMP}$-containing species such as *Providencia rettgeri*, *Providencia stuartii*, *Morganella morganii* and *Proteus mirabilis* do not have well-defined MLST schema. As such, we employed a two-tiered approach to classify lineages for this study. Initially, untyped species were assigned using MASH v2.3[68] into clusters if they were within ≤0.05 of each other to accurately group independent species, then PopPUNK v4.2.0[69] was used on each of these MASH clusters to assign lineages for within-species comparisons. The 'create-db' function was used with the following options: '--sketch-size 1000000 --min-k 15 --max-k 29 --qc-filter prune'. Then the 'fit-model' function was used with the following options: 'bgmm --ranks 1,2,3,5 --graph-weights'. Various --K values were used to obtain the best-scoring model fitting. Additionally, the 'poppunk_visualise' function was used, with the '--distances' and '--previous-clustering' to output a neighbour-joining core tree.

We identified that species identified as *Providencia rettgeri* and *Providencia stuartii* were considerably divergent and the taxonomy of these species requires detailed revision, with five and two independent species groupings outside the 0.05 standard threshold. These displayed MASH distances of up to 0.19 'within species', indicating that although these 'species' are related, they are distinct species or potentially genera. However, as this is beyond the scope of our current work, we refer to these as *Providencia rettgeri* spp. 1, *Providencia rettgeri* spp. 2, etc.

For assemblies without recorded sequencing technology, a heuristic approach was adopted: assemblies with fewer than 20 contigs were classified as long-read or hybrid, while those with 20 or more contigs were marked as short-read. We assumed that long-read sequencing or assemblies with both short- and long-read data typically yields fewer contigs compared to short-read sequencing data alone. The United Nations geoscheme was used to classify countries into regions and subregions. This resulted in a dataset comprising 536 long and 4022 short read assemblies.

### Assembly dereplication and IMP-cluster definition

To determine the impact of potential clonal outbreaks, we used a genomically-dereplicated dataset where a single representative of genetically similar genomes was selected. We did this by analysing pairwise SNV distances between genomes within the same ST/CG using the 'fasta' command from SKA v1.0.0[70], followed by the 'distance' command. Genomes were considered genetically similar if two genomes had ≤5 pairwise SNVs per Mb. We performed graph-based clustering using pairwise SNV differences as edges in igraph v2.0.3[71]. 'IMP-clusters' were defined as genomes which shared the same $bla_{IMP}$ variant, species, ST/CG, and were within the ≤5 pairwise SNVs per Mb threshold.

### Plasmid analysis

Plasmids were identified using mob_recon from MOB-suite v3.1.8[25], assigning contigs to plasmids and identifying replicon types. This was performed on long-read assemblies only, resulting in 433 plasmid contigs. These 433 plasmid-positive contigs were used to construct a custom $bla_{IMP}$-positive plasmid database via the 'mob_typer --multi', then 'mob_cluster --mode build' commands. Short-read assemblies were then screened against this custom database with mob_recon.

### Genetic context of mobile genetic elements

Flanker v0.1.5[72] was used to determine genetic context of $bla_{IMP}$ genes with the following commands: '--flank both --window 10000 --gene blaIMP --include_gene --cluster'. For analysis of mobile elements, only sliced contigs with ≥6 kb were used. Integrons were identified using Integron_finder v2.0.5[73] with the '--local-max --func-annot --union-integrases' options. IS were screened using ISEScan v1.7.2.3[74]. ICEs were identified using the ICEberg 2.0[75] database as a query against each genome with minimap2 v2.26[65].

### Protein and structural analysis

SignalP v6.0[76] was used to process the signal sequences from IMP protein sequences, then Clustal Omega v1.2.4[77] was used to align these sequences. Multiple Sequence Alignment tree was annotated with TreeBranchLabeller at https://github.com/bananabenana/TreeBranchLabeller. Multiple Sequence Alignment was also used as input to calculate amino acid conservation scores using the conserve() command from bio3d v2.4-4[78], with the following parameters: 'method = 'similarity', sub.matrix = 'bio3d''. Colabfold v1.5.5[79] was used to generate AlphaFold2[80] structure predictions of mature IMP sequences, with the rank 1 structures kept. All structures were used as input for FoldMason v1.763a428[81] to generate per-residue structural conservation lDDT scores. Scores were used to colour structures by residue using ChimeraX v1.8[82] and custom scripts at https://github.com/bananabenana/ChimeraX_scripts[83].

### Statistical analysis

Statistics were performed in R v4.4.1[84] and RStudio v2024.09.0[85]. The following R packages were used: tidyverse v2.0.0[86], colorspace v2.1-1[87], viridis v0.6.5[37], ggh4x v0.2.8[88], ggstream v0.1.0[89], maps v3.4.2[90], scatterpie 0.2.4[91], sf v 1.0-17[92], rnaturalearth v1.0.1[93], ggnewscale v0.5.0[94], treemapify v2.5.6[95], patchwork v1.3.0[96], igraph v2.0.3[71],

qgraph v1.9.8[97], ggraph v2.2.1[98], ggforce v0.5.0[99], ggalluvial v0.12.5[100], ggtree v3.12.0[101], treeio v1.28.0[102] and aplot v0.2.3[103]. All R code can be found at Figshare: https://doi.org/10.6084/m9.figshare.28440992.

## Reporting summary

Further information on research design is available in the Nature Portfolio Reporting Summary linked to this article.

## Data availability

All data generated in this study are provided in the Supplementary Data 1–11, Supplementary Information and Source Data files. Additional supplementary data is available at Figshare: https://doi.org/10.6084/m9.figshare.28440992. Source data are provided with this paper.

## Code availability

All analysis code is available at Figshare: https://doi.org/10.6084/m9.figshare.28440992. Custom scripts can also be found at https://github.com/bananabenana/residue_structure_colour_scripts and https://github.com/bananabenana/TreeBranchLabeller.

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

## Acknowledgements

This work was supported by an NHMRC Emerging Leader 1 Fellowship (APP1176324) awarded to N.M., NHMRC Practitioner Fellowship (APP1117940) awarded to A.Y.P., and NIH/NIAID R01AI175414 awarded to A.G-S. This research/work was supported by Monash eResearch capabilities, including M3 and Research Data Storage.

## Author contributions

Conceptualization: N.M. and A.Y.P., Data Curation: B.R.M. and B.V., Formal Analysis: B.R.M., B.V. and H.A.N., Funding Acquisition: N.M. and A.Y.P., Methodology: B.V., B.R.M. and N.M., Project Administration: N.M. and A.Y.P., Resources: N.M. and A.Y.P., Supervision: N.M. and A.Y.P., Writing – Original Draft Preparation: B.R.M., B.V. and N.M., Writing – Review & Editing: B.R.M., B.V., H.A.N., A.G-S., A.Y.P., NM, Both first-authors (B.V. and B.R.M.) can present as first-author on paper for grants and resumes.

## Competing interests

The authors declare no competing interests.
