## [Peer Review file · Nature Communications]

The rise and global spread of IMP carbapenemases (1996-2023): a genomic epidemiology study

Corresponding Author: Dr Nenad Macesic

Version 0:

Reviewer comments:

Reviewer #1

(Remarks to the Author)

This manuscript presents a comprehensive genomic epidemiology study of the blaIMP gene, leveraging 4,556 genomes collected over three decades across the globe. The authors trace the dissemination of blaIMP across 93 different bacterial species and provide a detailed breakdown by species. The study proceeds with an in-depth analysis of associated plasmids and mobile genetic elements, structural comparisons of IMP variants via protein modelling, and an exploration of the sources of the sequenced samples. Overall, this is an important and timely contribution to the field, offering valuable insights into the evolution and spread of blaIMP. I also commend the excellent use of graphics to illustrate the results.

However, I would like to raise several points for clarification and improvement:

1. Genome Data Quality While the authors state that assembly quality was assessed using QUAST, no summary of the results is provided. This leaves readers assuming that all genomes are of high quality, which may not be the case. I recommend including a brief summary of relevant quality metrics (e.g., N50, number of contigs, genome completeness) to support the reliability of the dataset.
2. Gene-Based Phylogenetic Tree Although the manuscript presents an amino acid-based phylogenetic tree of IMP variants in Figure 6, a nucleotide-based phylogeny using blaIMP sequences would be important in a genomic epidemiology context at the start of the analysis. Including such a comparison would enhance the biological relevance of the variant classification and better contextualize downstream analyses.
3. Whole-Genome Phylogenetic Tree A whole-genome phylogeny could help clarify whether blaIMP dissemination is due to clonal expansion or horizontal transfer across different lineages. Overlaying metadata such as host species, collection date, and plasmid types may provide additional picture of the gene's evolutionary and epidemiological dynamics.
4. AlphaFold Model Quality The manuscript would benefit from a brief comment on the quality of the AlphaFold structural models used. Were confidence metrics such as pLDDT scores used to validate model reliability?
5. Visualisation of Key Residues Figure 6 would be more informative if it included annotations of key residues in the protein structure. This would help readers relate amino acid conservation to structural context, particularly in regions of low conservation that may correspond to key residues.
6. Functional Implications of Mutations The discussion could be strengthened by elaborating on the functional implications of specific mutations reported in previous studies.

For transparency per editorial request, I used ChatGPT to polish the original version of the review (to amend the English in the report). I confirm that the manuscript was not uploaded into ChatGPT. I fully stands behind all statements written in my review.

(Remarks on code availability)

Looks okay.

Reviewer #2

(Remarks to the Author)

Thanks for asking me to review this really interesting study and manuscript by Vezina et al, which fills an important knowledge gap with respect to our understanding of the global genomic epidemiology of one of the five major clinically relevant carbapenemase genes, blaIMP. One of the things that I really like about this study is the focus on the gene (and

gene variant) as the unit of analysis, considering dissemination in the context of a genetic hierarchy (within mobile contexts, lineages and across species), across multiple One Health domains, and across geography and over time. The consideration of the impact of mutations on hydrolytic capacity as part of the story is also great. In addition I like the fact that the authors have clearly considered the caveats of the sampling frame (i.e. biases in publicly available sequencing datasets, the contribution of oversampling within outbreaks prompting an evaluation of the data post-deduplication).

To my mind this sort of multi-dimensional framework is key to giving us the best opportunity to understand how specific important gene variants emerge and what facilitates their spread over time, although the analysis presented here is largely descriptive and understanding the exact combination of factors that contribute to this emergence and dissemination remains elusive.

The manuscript is well-written and the figures are very nice.

For the figshare repository could the authors please include the full figshare doi in the manuscript:

<https://doi.org/10.6084/m9.figshare.28440992>

The R markdown is great but ideally the authors would also please provide the input files that are relevant to running the R markdown in this repository? Sorry if I have missed this.

Major comments/queries:

1. In characterising the IMP gene variants present in the data, could I just confirm that there was a requirement for a 100% nucleotide-level match for a specific blaIMP variant to be called as present? The thresholds for AMR gene assignment are not described clearly in the methods: We then performed resistance gene detection with AMRFinderPlus v3.12.8.

2. What's the predominant genetic context for blaIMP in *Pseudomonas aeruginosa*? It doesn't seem to be represented in Fig.4A much at all, so is it chromosomal, or just not able to be characterised given the limitations of short-read data? What do the authors think it is about *Pseudomonas aeruginosa* and blaIMP that have led to such a tight relationship – including for multiple different IMP variants – is there anything about where blaIMP is found in the *P. aeruginosa* genomes that might explain this? Was there anything else interesting to be said about the ~20% of cases that had chromosomal blaIMP given that chromosomal integration for other important AMR genes such as CTX-M-15 is increasingly described and may facilitate stable inheritance and persistence within populations?

3. Is there something about plasmid populations i.e. the other plasmids found in association with blaIMP-plasmids that contribute to IMP-plasmid associated dissemination/persistence? This would be interesting to analyse in the long-read data/fully reconstructed assemblies.

4. Perhaps I have missed it, but I couldn't see anywhere that the authors had considered the possibility that blaIMP-26 has emerged from blaIMP-4 given it differs by a single nucleotide and has previously been described as sharing similar genetic contexts to blaIMP-4 in some settings e.g. <https://pmc.ncbi.nlm.nih.gov/articles/PMC6629617/#CR12>. Was this hypothesis supported by the large-scale genomic analysis here?

5. I couldn't fully reconcile some of the numbers describing the different sequencing datasets being considered. The genome denominator seems to be variably 4,559 or 4,556 – please check this carefully throughout the text and e.g. in Table S1 where 4559 genomes are listed. Also in Lines 198-201 where short-read assemblies are then clustered to the long-read derived reference plasmid set I couldn't work out why the denominator was 4,559? Shouldn't this be 4,020 (i.e. considering only 4,020 short-read genomes described in L68?)

In Table S7, should there be a "short-read" category in addition to the "long-read/hybrid" category represented in the "assembly_type" column?

6. Fig.4 – Could the authors include a panel where the nodes are coloured by IMP variant e.g. using the same classification in Fig.S1 (i.e. the 8 major IMP variants individually and the other 46 variants as a group?) I think this visual representation would potentially help in understanding the distribution of IMP genes within plasmids?

7. L233-234 – "We analysed the genetic context 10kb up- and downstream of each blaIMP gene. To maximise identification of mobile elements, only contigs ≥ 6 kb in length were considered for this analysis which resulted in 2,314/4,559 eligible blaIMP-containing contigs (Fig. 5)." I don't fully understand how it is possible to analyse 10kb up and down stream of blaIMP if contigs <20kb were being considered? Please rephrase or clarify this statement? I also think it would be important to be explicit that this approach potentially results in a biased representation of the flanking sequences around these genes. Assemblies commonly fragment due to the presence of ISs/smaller mobile elements as these are often found in multiple copies within genomes. I would be a bit more cautious in the interpretation here given that only 51% of the data were evaluable?

8. Fig.6B – The legend suggests this shows: "Predicted AlphaFold2 structures of each 52 IMP variants, structurally aligned." – to me this just looks like a single structure? Would it be worthwhile annotating this figure with the catalytic site and the position of the residues highlighted in the heatmap instead? Also - I am not able to access any alignment at "Figshare: 10.6084/m9.figshare.28440992." – please check this is available.

9. Line 275 – "Non-human reservoirs..."

I think here it would be critically important to understand what the extent of One Health overlap is for IMP gene variants with shared genetic contexts (i.e. flanks/plasmids), and not just the overlap at the genome-level?

Minor queries:

10. Lines 175-176 - Were there any isolates that had more than one IMP variant or multiple IMP variants, or an IMP gene found on both the chromosome and a plasmid, or was IMP always present in a single copy in the long read/hybrid assemblies?

11. Lines 443-444 – “This resulted in a dataset comprising 536 long and 4,022 short read assemblies”

Lines 456-457 – “These 433 plasmid-positive contigs were used to construct a custom blaIMP-positive plasmid database...”

Lines 174-176 – “To determine the genetic context of blaIMP genes, we first analysed all available long-read genomes in the dataset (n=539) (see Methods). blaIMP genes were located on plasmids in 436/539 genomes (80.9%).”

These seem to suggest slightly different numbers of long-read/hybrid assemblies (536 vs 539) and blaIMP-positive plasmids (436 versus 433) that were observed in these assemblies – please review, reconcile and correct.

12. Line 456 – suggests 433 IMP-positive plasmid contigs in long-read/hybrid assemblies –

13. Table S7 – I couldn't understand why if you sum the number_of_occurrences_in_the_dataset column this adds up to 2723?

14. There are no statistical evaluations made in support of some of the differences identified between groups of observations – could the authors comment on why they have chosen not to undertake these?

Syntax/typographical suggestions:

Line 71 – delete total to give : “...totalling 4,559 total blaIMP genes...”

Fig.2C legend – C: should read “Global distribution of blaIMP-27...” I think?

(Remarks on code availability)

The R markdown looks good on a quick skim, but I have not been able to run it as the input files have not been provided (at least that I can see).

Reviewer #3

(Remarks to the Author)

In my opinion this is a well designed and executed study which is presented clearly. The limitations of bias from use of publicly available sequencing data and lack of high quality long read assemblies are appropriately acknowledged.

The authors have done an impressive job of collating strains – there are 4319 strains in AllTheBacteria with blaIMP vs their 4556. I am reassured that they have used a comprehensive and effective search strategy.

The methods are generally well written and clear and the code has been made available. The figures are clear and high quality.

I have only some relatively minor comments for the authors consideration:

1) Please clarify what you mean by a “de-replicated” dataset for cluster definition (L446). Presumably you didn't have patient level metadata, so I'm not sure how you can actually do this. There is a risk that sequencing of multiple isolates from the same patient over-inflates the size of some clusters.

2) Where all instances of blaIMP genes found at 100% identity and coverage matches to a reference allele? Please add this information. What did you do with non-exact matches if these occurred? Note that AMRFinderPlus won't tell you about synonymous mutations in these genes. I'm slightly unclear if the section from L154 refers to the already catalogued IMP variants or whether you also looked for uncatalogued additional mutations in these? This comment is also relevant to other sections where you describe the total number of IMP alleles observed.

3) The authors propose a real-time coordinated (presumably WGS informed) public health response to blaIMP dissemination. To my knowledge, whilst more surveillance is appealing, there is no evidence that it is effective in reducing the incidence of AMR outside of very specific localised outbreaks in healthcare settings.

Minor

L455-456 – please rephrase this doesn't make sense.

(Remarks on code availability)

The code is well annotated and looks appropriate for the analysis presented.

Version 1:

Reviewer comments:

Reviewer #1

(Remarks to the Author)

All of my concerns have been addressed.

(Remarks on code availability)

Reviewer #2

(Remarks to the Author)

Thanks for addressing my comments and including the additional analyses where suggested - I have no further comments.

(Remarks on code availability)

I have not reviewed this again.

Dear Reviewers,

Thank you for your careful consideration of our manuscript '*The rise and global spread of IMP carbapenemases (1996-2023): a genomic epidemiology study*'. Your comments are much appreciated, and we believe your input has strengthened the manuscript. Please find our point-by-point response to your comments below. All line numbers refer to the manuscript with tracked changes (Simple Markup mode).

REVIEWER #1

Comment 1:

Genome Data Quality: While the authors state that assembly quality was assessed using QUAST, no summary of the results is provided. This leaves readers assuming that all genomes are of high quality, which may not be the case. I recommend including a brief summary of relevant quality metrics (e.g., N50, number of contigs, genome completeness) to support the reliability of the dataset

Response 1:

Line 459-461 has been updated to reflect our quality control thresholds. The QUAST results have been included in Table S1:

“Assembly quality was checked using Quast v5.3.0 (56), and all genomes with an N50 \geq 10,000 and \leq 1000 contigs were included (**Table S1**).”

Comment 2:

Gene-Based Phylogenetic Tree: Although the manuscript presents an amino acid-based phylogenetic tree of IMP variants in Figure 6, a nucleotide-based phylogeny using blaIMP sequences would be important in a genomic epidemiology context at the start of the analysis. Including such a comparison would enhance the biological relevance of the variant classification and better contextualize downstream analyses.

Response 2:

Thank you for the suggestion. We feel that this analysis may not provide valuable information for the manuscript as the most biologically relevant characteristics of IMP variants are their protein sequences and their variable catalytic activity against carbapenem antibiotics. Specifically, analysis in the protein space allowed us to remove the signal sequence from the IMP proteins, which would otherwise introduce noise into alignments. For example, IMP-26 and IMP-89 are considered different variants, yet they differ by a N14S mutation within the signal sequence region of the protein. For the mature protein, these two variants are functionally identical, which can be seen in Fig. 6. A DNA alignment indicates a single bp mutation in position 41 from A>G. Therefore while we agree that DNA alignments are useful, analysis in the protein space contextualises mutations as biologically-relevant residue changes.

We have expanded the discussion to talk about the biological implications of these mutations at lines 373-381:

“Beyond blaIMP gene transfer, we also analysed the structures of IMP enzymes to examine whether adaptive changes themselves may be contributing to spread. Despite the diversity of 52 known IMP variants, we detected convergent evolutionary patterns, with repeated missense mutations at specific sites. While not every residue across each variant has been studied within the context of AMR, these changes likely offer functional advantages by altering carbapenem hydrolytic activity and thus minimum inhibitory concentrations, as previously seen for V31F, S196G and N167Y (30,33–35,49). Many of these convergent mutations appear to have been acquired independently, such as 31F found in divergent blaIMP-10 and blaIMP-26 sequences (Fig. 6) indicating a common evolutionary solution that allows the successful proliferation of IMP variants.”

Comment 3:

Whole-Genome Phylogenetic Tree: A whole-genome phylogeny could help clarify whether blaIMP dissemination is due to clonal expansion or horizontal transfer across different lineages. Overlaying metadata such as host species, collection date, and plasmid types may provide additional picture of the gene's evolutionary and epidemiological dynamics.

Response 3:

Thank you for the suggestion. Unfortunately, a whole genome phylogeny of 93 bacterial species will likely not provide clarity. In our current manuscript, we have already attempted to classify if IMP dissemination was due to clonal expansion or plasmid transmission (see Figure 4). By firstly splitting genomes into species, then Clonal Groups, then SNP clusters, we could effectively de-replicate clonally-linked genomes from each other. This reduced dataset of representative IMP-clusters was then assessed for plasmid transmission. Please see Lines 149-164 for details and Lines 499-500 for methods.

Comment 4:

AlphaFold Model Quality: The manuscript would benefit from a brief comment on the quality of the AlphaFold structural models used. Were confidence metrics such as pLDDT scores used to validate model reliability?

Response 4:

Thank you for the suggestion.

This information has been added to Lines 280-284 and to Table S9:

“Predicted structures for each mature IMP variant were found have high confidence, with a mean predicted local distance difference test (pLDDT) of 96.36 ± 7.23 SD across all residues and variants (**Table S9**). When comparing IMP variant structures to each other, we found they were highly structurally conserved, with a mean IDDT score of 0.977.”

Comment 5:

Visualisation of Key Residues: Figure 6 would be more informative if it included annotations of key residues in the protein structure. This would help readers relate amino acid conservation to structural context, particularly in regions of low conservation that may correspond to key residues.

Response 5:

Fig. 6 has now been updated to show these key residues.

Comment 6:

Functional Implications of Mutations: The discussion could be strengthened by elaborating on the functional implications of specific mutations reported in previous studies.

Response 6:

Information has been added in Lines 373-383:

“Beyond blaIMP gene transfer, we also analysed the structures of IMP enzymes to examine whether adaptive changes themselves may be contributing to spread. Despite the diversity of 52 known IMP variants, we detected convergent evolutionary patterns, with repeated missense mutations at specific sites. While not every residue across each variant has been studied within the context of AMR, these changes likely offer functional advantages by altering carbapenem hydrolytic activity and thus minimum inhibitory concentrations, as previously seen for V31F, S196G and N167Y (30,33–35,49). Many of these convergent mutations appear to have been acquired independently, such as 31F found in divergent blaIMP-10 and blaIMP-26 sequences (Fig. 6) indicating a common evolutionary solution that allows the successful proliferation of IMP variants. We identified IMP-26, as a key regional endemic clone in Asia, which is more effective at hydrolysing meropenem and doripenem than older variants and thus displays a broader and more effective carbapenemase phenotype (53).”

REVIEWER #2

Comment 1:

For the figshare repository could the authors please include the full figshare doi in the manuscript: <https://doi.org/10.6084/m9.figshare.28440992>

Response 1:

This has now been updated at every location within the manuscript (Lines 298, 532, 537).

Comment 2:

The R markdown is great but ideally the authors would also please provide the input files that are relevant to running the R markdown in this repository? Sorry if I have missed this.

Response 2:

This has now been uploaded to Figshare as a zipped R_markdown_input_data.7z. To use, unzip and execute R code within directory. This can be found at <https://doi.org/10.6084/m9.figshare.28440992>.

Comment 3:

In characterising the IMP gene variants present in the data, could I just confirm that there was a requirement for a 100% nucleotide-level match for a specific blaIMP variant to be called as present? The thresholds for AMR gene assignment are not described clearly in the methods: We then performed resistance gene detection with AMRFinderPlus v3.12.8.

Response 3:

We have clarified this in Lines 463-464:

“IMP variants were only considered if they had a 100% match for both identity and query coverage (Table S1). In one case (SRR17656613), a partial match was found, which was confirmed via minimap2 v2.26 (71) alignment of blaIMP alleles to contigs linked within the assembly graph.”

Comment 4:

What's the predominant genetic context for blaIMP in *Pseudomonas aeruginosa*? It doesn't seem to be represented in Fig.4A much at all, so is it chromosomal, or just not able to be characterised given the limitations of short-read data? What do the authors think it is about *Pseudomonas aeruginosa* and blaIMP that have led to such a tight relationship – including for multiple different IMP variants – is there anything about where blaIMP is found in the *P. aeruginosa* genomes that might explain this? Was there anything else interesting to be said about the ~20% of cases that had chromosomal blaIMP given that chromosomal integration for other important AMR genes such as CTX-M-15 is increasingly described and may facilitate stable inheritance and persistence within populations?

Response 4:

Thank you for this insightful comment. Among 977 *P. aeruginosa* genomes, blaIMP was chromosomally located in 59 genomes, which is the highest number of chromosomal integrations observed for any species in our dataset. An additional 199 genomes carried blaIMP on plasmids, while 719 were “unclassified” due to the absence of known replicon types. We did not speculate on these unclassified cases, since many replicon types remain undefined. There was, however, a striking lineage association, as you noted. For example, blaIMP-26 was tightly linked with ST235: 136 genomes carried it on unclassified molecules, and 22 on plasmids (20/22 with unknown replicon types). We have updated the manuscript accordingly/

Lines 232–236:

“*P. aeruginosa* was notable for having the greatest number of chromosomally located blaIMP genes of all species (n=59). Of the 977 genomes, 199 harboured plasmid-borne blaIMP, and 719 were ‘unclassified’ due to lack of replicon typing. We did not speculate on the unclassified group. Nonetheless, IMP variants showed strong lineage coupling, such as blaIMP-26 in ST235, where 136 genomes carried it on unclassified molecules and 22 on plasmids (20/22 with unknown replicon types).”

Lines 365-369:

“The chromosomal integration of blaIMP in *P. aeruginosa* likely facilitates stable inheritance and persistence within successful clones, supporting ongoing clonal spread. This is consistent with reports of chromosomal integration of other key resistance genes, such as blaCTX-M-15, which similarly enhances stability and long-term maintenance within bacterial populations (51).”

Comment 5:

Is there something about plasmid populations i.e. the other plasmids found in association with blaIMP-plasmids that contribute to IMP-plasmid associated dissemination/persistence? This would be interesting to analyse in the long-read data/fully reconstructed assemblies.

Response 5:

We have now generated a network and performed network-based degree analysis of the associations between IMP and non-IMP plasmids within genomes.

Lines 199-207:

“We then assessed co-occurrence of these 433 *bla*_{IMP}-plasmids and non-*bla*_{IMP} plasmids within the same genomes via network analysis. To account for clonal bias, single representatives from each IMP-cluster were used (**Fig. S3**). Analysis of node degrees (number of edges for each node) revealed that *bla*_{IMP-4} plasmids had the greatest number of non-*bla*_{IMP} plasmid co-associations within genomes. In particular *bla*_{IMP-4} IncC, *bla*_{IMP-4} IncL/M and *bla*_{IMP-4} IncN plasmids had the highest network degrees (37, 33 and 25, respectively). Plasmid co-occurrence and dissemination may be shaped by both bacterial host traits (*E. hormaechei*, *E. coli* and *K. pneumoniae*) and plasmid-specific factors. Enterobacterales are known for their ability to carry multiple plasmids (26) and were noted to have high centrality in our data, suggesting that host factors play the defining role in this co-occurrence analysis.”

Comment 6:

Perhaps I have missed it, but I couldn't see anywhere that the authors had considered the possibility that *bla*_{IMP-26} has emerged from *bla*_{IMP-4} given it differs by a single nucleotide and has previously been described as sharing similar genetic contexts to *bla*_{IMP-4} in some settings e.g. <https://pmc.ncbi.nlm.nih.gov/articles/PMC6629617/#CR12>. Was this hypothesis supported by the large-scale genomic analysis here?

Response 6:

Thank you for bringing this to our attention. This reference has now been included and we have addressed this issue in the Discussion.

Lines 383-388:

“It is reasonable to speculate that *bla*_{IMP-26} evolved from *bla*_{IMP-4}, given their single nucleotide polymorphism difference while sharing similar genetic contexts (53). Indeed, our dataset shows *bla*_{IMP-4} was first noted genomically in 1998, followed by *bla*_{IMP-26} in 2009. While *bla*_{IMP-4} outnumbered *bla*_{IMP-26} (n=1,592 vs n=254), they share overlap of six geographical subregions and 13/14 species carrying *bla*_{IMP-26} can also carry *bla*_{IMP-4}, raising the possibility that there have been multiple independent *bla*_{IMP-26}-evolution events.”

Comment 7:

I couldn't fully reconcile some of the numbers describing the different sequencing datasets being considered. The genome denominator seems to be variably 4,559 or 4,556 – please check this carefully throughout the text and e.g. in Table S1 where 4559 genomes are listed. Also in Lines 198-201 where short-read assemblies are then clustered to the long-read derived reference plasmid set I couldn't work out why the denominator was 4,559? Shouldn't this be 4,020 (i.e. considering only 4,020 short-read genomes described in L68?)

Response 7:

4,556 genomes were analysed, with 4,559 *bla*_{IMP} genes detected due to dual carriage in 3 genomes. This is shown in Table S1 where each row shows an IMP gene instance, with 3 genomes listed twice. We have corrected this to 4,556 genomes in each case.

Lines 68-71 explicitly state:

“We identified 4,556 genomes (4,020 assembled from short- and 536 from long-read sequencing data) isolated globally from 1996-2023 carrying 52 distinct *bla*_{IMP} variants across 26 bacterial genera (Fig. 1 and Table S1). This revealed a remarkable diversity of both *bla*_{IMP} genes and their bacterial hosts, totalling 4,559 *bla*_{IMP} genes, with three long-read genomes carrying two *bla*_{IMP} variants each.”

We have also updated lines 202-204 to reflect the results of the entire dataset rather than just short read assemblies.

Lines 210-213:

“Across the entire short and long-read dataset, *bla*_{IMP} variants were found on plasmids in 2,909/4,556 (63.8%) genomes, the chromosome in 97 genomes, with 1,553 remaining unclassified (Table S1).”

Comment 8:

In Table S7, should there be a “short-read” category in addition to the “long-read/hybrid” category represented in the “assembly_type” column?

Response 8:

Table S7 was specifically referring to the long-read-only analysis. We have updated Line 202 to reference Table S1 rather than S7.

Comment 9:

Fig.4 – Could the authors include a panel where the nodes are coloured by IMP variant e.g. using the same classification in Fig.S1 (i.e. the 8 major IMP variants individually and the other 46 variants as a group?) I think this visual representation would potentially help in understanding the distribution of IMP genes within plasmids?

Response 9:

Fig. 4 has been updated to include panel C, coloured by IMP variant.

Comment 10:

L233-234 – “We analysed the genetic context 10kb up- and downstream of each *bla*_{IMP} gene. To maximise identification of mobile elements, only contigs ≥ 6 kb in length were considered for this analysis which resulted in 2,314/4,559 eligible *bla*_{IMP}-containing contigs (Fig. 5).” I don’t fully understand how it is possible to analyse 10kb up and down stream of *bla*_{IMP} if contigs < 20 kb were being considered? Please rephrase or clarify this statement? I also think it would be important to be explicit that this approach potentially results in a biased representation of the flanking sequences around these genes. Assemblies commonly fragment due to the presence of ISs/smaller mobile elements as these are often found in multiple copies within genomes. I would be a bit more cautious in the interpretation here given that only 51% of the data were evaluable?

Response 10:

As indicated by the reviewer, only 51% of the data was usable due to the shortcomings of plasmid reconstruction using short-read assembly. We have addressed this in the manuscript.

Line 248-252 now read:

“We analysed the genetic context up to 10 kb up- and down-stream of each *bla*_{IMP} gene. To maximise identification of mobile elements and analysis dataset, only contigs ≥ 6 kb in length were considered for this analysis which resulted in 2,314/4,559 eligible *bla*_{IMP}-containing contigs (Fig. 5). A threshold of ≥ 6 kb was used as this was a natural cutoff point when analysing the distribution of *bla*_{IMP}-containing contig lengths (data not shown).”

Lines 257-259:

“This analysis was limited by the lack of eligible contigs due to the dataset being comprised mostly of short-read assemblies. Contig breaks are commonly caused by the presence of IS and repetitive elements (31), potentially biasing *bla*_{IMP} flanking region analyses. However, we consistently found intact IS within these flanking regions, showing the validity of this approach.”

Comment 11:

Fig.6B – The legend suggests this shows: “Predicted AlphaFold2 structures of each 52 IMP variants, structurally aligned.” – to me this just looks like a single structure? Would it be worthwhile annotating this figure with the catalytic site and the position of the residues highlighted in the heatmap instead? Also - I am not able to access any alignment at “Figshare: 10.6084/m9.figshare.28440992.” – please check this is available.

Response 11:

Fig 6B shows overlays of all 52 IMP variants – the structures are highly conserved, as seen in the high IDDT scores in Fig 6C which is also shown on a single structure. This has been updated to include the residues highlighted in the heatmap.

Figshare link has also been correctly updated within the manuscript:
<https://doi.org/10.6084/m9.figshare.28440992>

Comment 12:

Line 275 – “Non-human reservoirs...” I think here it would be critically important to understand what the extent of One Health overlap is for IMP gene variants with shared genetic contexts (i.e. flanks/plasmids), and not just the overlap at the genome-level?

Response 12.

Thank you for the suggestion. We have now analysed this and updated the manuscript.

Lines 311-317:

“We then examined the transmission of plasmids between sources, after accounting for IMP-clusters. We identified at least 20 independent plasmids moving between sources, most commonly between clinical and environmental categories (15/20). These included *bla*_{IMP-4} (n=15), *bla*_{IMP-6} (n=2), *bla*_{IMP-1} (n=1) and *bla*_{IMP-8} (n=1) (Table S11). *bla*_{IMP-4} IncN plasmids were the most notable, spreading between clinical and environmental source categories via 36 individual clones across eight species. We also found one case of a *bla*_{IMP-4} IncHI2A plasmid moving between clinical, environmental and animal source categories across 13 genomes from six species.”

Comment 13:

Lines 175-176 - Were there any isolates that had more than one IMP variant or multiple IMP variants, or an IMP gene found on both the chromosome and a plasmid, or was IMP always present in a single copy in the long read/hybrid assemblies?

Response 13:

We have addressed this in lines 183-184:

Three long-read genomes carried two IMP variants, each co-located on the same molecule (Unknown rep plasmid, chromosomally, and IncFIA:IncFIB, respectively) (Table S1).

Comment 14:

Lines 491-492 – “This resulted in a dataset comprising 536 long and 4,022 short read assemblies”.
Lines 504-506 – “These 433 plasmid-positive contigs were used to construct a custom *bla*_{IMP}-positive plasmid database...”.

Lines 175-177 – “To determine the genetic context of *bla*_{IMP} genes, we first analysed all available long-read genomes in the dataset (n=539) (see Methods). *bla*_{IMP} genes were located on plasmids in 436/539 genomes (80.9%)”. Line 456 – suggests 433 IMP-positive plasmid contigs in long-read/hybrid assemblies. These seem to suggest slightly different numbers of long-read/hybrid assemblies (536 vs 539) and *bla*_{IMP}-positive plasmids (436 versus 433) that were observed in these assemblies – please review, reconcile and correct.

Response 14:

Thank you for identifying these discrepancies. Regarding plasmids, there were 433 IMP plasmids from 536 long read assemblies. Three of these assemblies contained an additional IMP variant (2x IMPs within genome), resulting in miscounts of 539 genomes and 436 plasmids.

Lines 70-71 were updated for clarity:

“This revealed a remarkable diversity of both *bla*_{IMP} genes and their bacterial hosts, totalling 4,559 *bla*_{IMP} genes, with three long-read genomes carrying two *bla*_{IMP} variants each.”

Lines 175-177 were also updated:

“To determine the genetic context of *bla*_{IMP} genes, we first analysed all available long-read genomes in the dataset (n=536) (see **Methods**). *bla*_{IMP} genes were located on plasmids in 433/536 genomes (81.3%).”

Lines 502-506 were re-phrased:

“Plasmids were identified using *mob_recon* from MOB-suite v3.1.8 (25), assigning contigs to plasmids and identifying replicon types. This was performed on long-read assemblies only, resulting in 433 plasmid contigs. These 433 plasmid-positive contigs were used to construct a custom *bla*_{IMP}-positive plasmid database via the “*mob_typer --multi*”, then “*mob_cluster --mode build*” commands. Short-read assemblies were then screened against this custom database with *mob_recon*.”

Comment 15

Table S7 – I couldn't understand why if you sum the `number_of_occurrences_in_the_dataset` column this adds up to 2723? There are no statistical evaluations made in support of some of the differences identified between groups of observations – could the authors comment on why they have chosen not to undertake these?

Response 15:

This column was erroneously calculated and included. It is now removed. This table intends to provide a description of plasmid distributions. Therefore, conducting a statistical analysis is beyond the intent of the table.

Comment 16:

Line 71 – delete total to give : “...totalling 4,559 total blaIMP genes...”

Response 16:

This has been removed.

Comment 17:

Fig.2C legend – C: should read “Global distribution of blaIMP-27...” I think?

Response 17:

This has been corrected.

Comment 18:

The R markdown looks good on a quick skim, but I have not been able to run it as the input files have not been provided (at least that I can see).

Response 18:

This has now been uploaded to Figshare as a zipped R_markdown_input_data.7z. To use, unzip and execute R code within directory. This can be found at <https://doi.org/10.6084/m9.figshare.28440992>.

REVIEWER #3

Comment 1:

Please clarify what you mean by a “de-replicated” dataset for cluster definition (L446). Presumably you didn’t have patient level metadata, so I’m not sure how you can actually do this. There is a risk that sequencing of multiple isolates from the same patient over-inflates the size of some clusters.

Response 1:

Thank you for this query. We de-replicated according to genetic similarity rather than patient metadata. As the reviewer noted, patient-level metadata were not available to us.

Lines 494-500 now read:

“To determine the impact of potential clonal outbreaks, we used a genomically-dereplicated dataset where a single representative of genetically similar genomes was selected. We did this by analysing pairwise SNV distances between genomes within the same ST/CG using the ‘fasta’ command from SKA v1.0.0 (65), followed by the ‘distance’ command. Genomes were considered genetically similar if two genomes had ≤ 5 pairwise SNVs per Mb. We performed graph-based clustering using pairwise SNV differences as edges in igraph v2.0.3 (66). ‘IMP-clusters’ were defined as genomes which shared the same *bla*_{IMP} variant, species, ST/CG, and were within the ≤ 5 pairwise SNVs per Mb threshold.”

Comment 2:

Were all instances of *bla*_{IMP} genes found at 100% identity and coverage matches to a reference allele? Please add this information. What did you do with non-exact matches if these occurred? Note that AMRFinderPlus won’t tell you about synonymous mutations in these genes. I’m slightly unclear if the section from L154 refers to the already catalogued IMP variants or whether you also looked for uncatalogued additional mutations in these? This comment is also relevant to other sections where you describe the total number of IMP alleles observed.

Response 2:

We have clarified this point in Lines 463-467:

“IMP variants were only considered if they had a 100% match for both identity and query coverage (Table S1). In one case (SRR17656613), a partial match was found, which was confirmed via minimap2 v2.26 (71) alignment of *bla*_{IMP} alleles to contigs linked within the assembly graph.”

We used previously catalogued IMP variants included in AMRFinderPlus. This has been updated in the manuscript.

Line 77 reads:

“We did not identify any uncatalogued *bla*_{IMP} variants.”

Comment 3:

The authors propose a real-time coordinated (presumably WGS informed) public health response to *bla*_{IMP} dissemination. To my knowledge, whilst more surveillance is appealing, there is no evidence that it is effective in reducing the incidence of AMR outside of very specific localised outbreaks in healthcare settings.

Response 3:

We agree that surveillance alone does not reduce AMR incidence. Impact depends on coupling surveillance to defined infection prevention and stewardship actions. Evidence from coordinated programs shows that when surveillance (including mandatory reporting and/or rapid sequencing) is linked to targeted interventions, transmission and cases can be reduced. For example, Israel’s national CRE intervention was built on active surveillance, mandatory reporting, and cohorting, and reversed a country-wide CRKP epidemic after local measures had failed (<https://pubmed.ncbi.nlm.nih.gov/21317398/>). We have tempered our language accordingly.

Lines 438-444:

‘Our work highlights the need for more robust and sophisticated surveillance approaches that address gaps in low- and middle-income countries, incorporate methodology to detect plasmid and mobile genetic element transmission and conduct more systematic sampling of One Health reservoirs. This enhanced surveillance needs to be linked to targeted infection prevention and antimicrobial stewardship activities to reduce IMP cases and transmission. We have carried out a systematic analysis of IMP carbapenemases using three decades of data but the critical threat they pose now requires a proactive, real-time and co-ordinated public health response.’

Comment 4:

L455-456 – please rephrase this doesn’t make sense.

Response 4:

Lines 502-506 were rephrased:

“Plasmids were identified using *mob_recon* from MOB-suite v3.1.8 (25), assigning contigs to plasmids and identifying replicon types. This was performed on long-read assemblies only, resulting in 433 plasmid contigs. These 433 plasmid-positive contigs were used to construct a custom *bla*_{IMP}-positive plasmid database via the “*mob_typer --multi*”, then “*mob_cluster --mode build*” commands. Short-read assemblies were then screened against this custom database with *mob_recon*.”